# CROSS-MODAL CONTENT OPTIMIZATION FOR STEERING WEB AGENT PREFERENCES

## ABSTRACT

Vision–language model (VLM)-based web agents increasingly power high-stakes selection tasks like content recommendation or product ranking by combining multimodal perception with preference reasoning. Recent studies reveal that these agents are vulnerable against attackers who can bias selection outcomes through preference manipulations using adversarial pop-ups, image perturbations, or content tweaks. Existing work, however, either assumes strong white-box access, with limited single-modal perturbations, or uses impractical settings. In this paper, we demonstrate, for the first time, that joint exploitation of visual and textual channels yields significantly more powerful preference manipulations under realistic attacker capabilities. We introduce Cross-Modal Preference Steering (CPS) that jointly optimizes imperceptible modifications to an item's visual and natural language descriptions, exploiting CLIP-transferable image perturbations and RLHF-induced linguistic biases to steer agent decisions. In contrast to prior studies that assume gradient access, or control over webpages, or agent memory, we adopt a realistic black-box threat setup: a non-privileged adversary can edit only their own listing's images and textual metadata, with no insight into the agent's model internals. We evaluate CPS on agents powered by state-of-the-art proprietary and open source VLMs including GPT-4.1, Qwen-2.5VL and Pixtral-Large on both movie selection and e-commerce tasks. Our results show that CPS is significantly more effective than leading baseline methods. For instance, our results show that CPS consistently outperforms baselines across all models while maintaining 70% lower detection rates, demonstrating both effectiveness and stealth. These findings highlight an urgent need for robust defenses as agentic systems play an increasingly consequential role in society.

## 1 INTRODUCTION

The architecture of human-AI interaction is undergoing a fundamental transformation. Traditional search engines and recommendation systems are rapidly giving way to AI-powered web agents that mediate an increasingly broad spectrum of digital experiences (Gartner, Inc., 2024). These agents extend far beyond simple information retrieval, serving as primary intermediaries for consequential decisions including product selection, job matching, content curation, and scholarly research recommendations.

This paradigm shift has prompted initial investigations into web agent security. Recent work has identified various attack vectors: HTML content injection (Xu et al., 2024; Liao et al., 2024), memory poisoning in retrieval-augmented systems (Zhong et al., 2023), and adversarial perturbations with white-box VLM access (Wu et al., 2024; Aichberger et al., 2025). While these studies highlight potential vulnerabilities, they rely on strong and often impractical assumptions—requiring either (1) full control over legitimate webpages (Xu et al., 2024; Liao et al., 2024), (2) write access to agent memory systems (Zhong et al., 2023), (3) prior knowledge of user queries (Zhang et al., 2025), or (4) white-box access to the underlying VLMs (Wu et al., 2024; Aichberger et al., 2025). Such requirements severely limit the practical application of these attacks in real-world deployments.

This work departs from these unrealistic assumptions to investigate a more practical and concerning threat model. We consider the common scenario where users delegate both search and selection tasks to AI agents. When multiple items match search criteria—whether products, services, or information

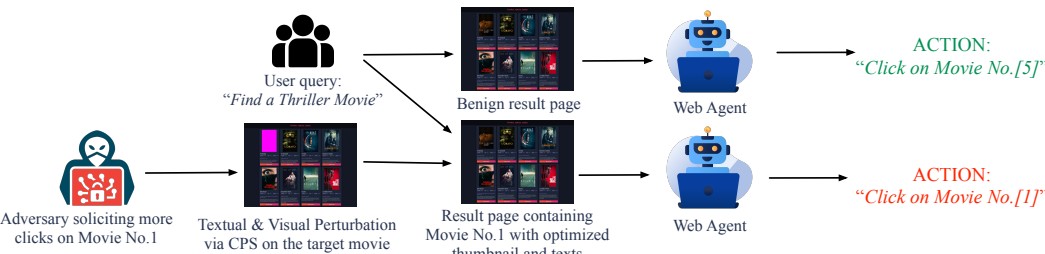

Figure 1: CPS attack on web agent selection. Top: benign scenario with random selection. Bottom: adversary jointly perturbs thumbnails and text to steer the agent toward selecting the targeted item.

sources—users increasingly rely on agents to make final selections. This delegation creates an automated decision pipeline with minimal human oversight, where adversarial manipulation can have direct economic and social consequences.

Our threat model reflects real-world constraints: attackers operate as content publishers with modification rights limited to their own content—specifically, item thumbnails and textual descriptions. They cannot alter webpage structure, access agent internals, or modify user queries. Despite these constraints, we demonstrate exploitation of two fundamental vulnerabilities in VLM-based web agents to systematically bias selection toward target items.

**Visual Vulnerability:** Prior work (Wu et al., 2024; Huang et al., 2025) identified that VLMs all rely on image encoders that share similar architecture and training data, creating a common attack surface. However, these methods suffered critical limitations: low success rates (below 20%), white-box access requirements, and mandatory image resizing to 224×224 pixels that hampered practical utility. We overcome these barriers through PGD-based optimization (Madry et al., 2019) against the embeddings of multiple augmented fragments of images through an ensemble of diverse CLIP architectures, achieving $> 50\%$ success rates against black-box commercial models (e.g., GPT-4.1) while preserving arbitrary image resolutions and visual imperceptibility.

**Textual Vulnerability:** Reinforcement Learning from Human Feedback (RLHF), while essential for alignment, inadvertently introduces exploitable behavioral biases (Bu et al., 2025; Kotek et al., 2023; Kumar et al., 2024). Models develop systematic preferences for specific response patterns and stylistic elements that adversaries can exploit through carefully crafted descriptions without triggering detection.

The synergy between visual and textual vulnerabilities presents an unprecedented security challenge. Our cross-modal approach operates through subtle, semantically plausible modifications that remain imperceptible to both automated systems and human observers while effectively steering agent behavior toward adversarial objectives.

## 1.1 OUR CONTRIBUTIONS

This work presents the first systematic investigation of web agent preference manipulation under realistic threat models. We introduce **Cross-Modal Preference Steering (CPS)**, a novel attack framework that synergistically exploits visual and textual vulnerabilities:

1. **Realistic Threat Model**: We formalize a practical attack scenario where adversaries operate as legitimate content publishers, exposing critical vulnerabilities affecting deployed web agents without requiring system access or webpage control.

2. **Black-Box Visual Attack**: We achieve >50% success rates in shifting VLMs' perceptions on images (e.g. apple $\leftrightarrow$ orange) through transferable adversarial examples from PGD (Madry et al., 2019), maintaining arbitrary image resolutions and visual imperceptibility, overcoming all limitations of prior work (Wu et al., 2024; Huang et al., 2025).

3. **Cross-Modal Framework**: We develop a unified methodology jointly optimizing visual perturbations and textual modifications, demonstrating that cross-modal coordination amplifies attack effectiveness beyond single-modal approaches.

Our findings reveal that AI agents' rapid deployment as digital intermediaries has created exploitable security gaps. The ability to invisibly steer preferences through imperceptible modifications, which is achievable by any content publisher, poses immediate risks to market fairness, user autonomy, and information integrity.

## 2 BACKGROUND & RELATED WORKS

In this section, we point out two key areas of research that are relevant to our work. We first discuss existing work that manipulates the behavior of Web / OS agents, followed by a discussion of RLHF that underlies a prominent weakness of LLMs that enable such attacks, which we target in our work.

### 2.1 ADVERSARIAL ATTACKS AGAINST WEB/OS AGENTS

Adversarial attacks on web agents can be categorized by their access requirements and threat assumptions. White-box attacks demonstrate severe vulnerabilities when granted model access: OS_Attack (Aichberger et al., 2025) and Agent_Attack (Wu et al., 2024) use PGD to perturb backgrounds or thumbnails with the goal of maximizing certain outputs from the white-box VLMs, embedding harmful instructions in parsed descriptions. However, such white-box access is rarely available in practice.

Black-box attacks require specific priors to succeed. Some exploit memory access: PoisoningRAG (Zhong et al., 2023), BadAgent (Wang et al., 2024), and PracticalInjection (Dong et al., 2025) inject poisoned information into agent memory systems. Others require knowledge of user inputs: PopupAttack (Zhang et al., 2025) adds adversarial pop-up boxes with "attention hooks" tailored to predicted queries. A third category assumes control over webpages themselves: EIA (Liao et al., 2024) injects instructions through form/mirror injection, while AdvAgent (Xu et al., 2024) inserts invisible prompts into non-rendered HTML fields. These webpage manipulation attacks, while effective, require unrealistic control over legitimate sites.

### 2.2 PREFERENCE MANIPULATION

RLHF, while aligning models with human values, inadvertently imprints systematic biases (Bu et al., 2025; Kotek et al., 2023; Kumar et al., 2024) such as verbosity preferences, sycophancy (Sharma et al., 2025) that create exploitable vulnerabilities for steering agent choices. Recent work demonstrates practical exploitation of these biases. Nestaas et al. craft subtle page text modifications that promote attackers in LLM search rankings without triggering detection (Nestaas et al., 2024). ToolHijacker (Shi et al., 2025) biases tool selection through optimized documents in retriever libraries. MPMA (Wang et al., 2025) achieves high selection rates using simple superlative descriptions ("best tool"), exploiting preferences for positive framing. These works collectively demonstrate that alignment-induced heuristics create a distinct attack surface for preference manipulation through seemingly benign textual modifications.

## 3 THREAT MODEL

We formalize the threat model for preference manipulation attacks against web agents by defining the task settings, attack surfaces, and evaluation metrics.

### 3.1 TASK SETTINGS

Consider a web agent $f_\theta : \mathcal{X} \to \mathcal{Y}$ that maps from input space $\mathcal{X}$ (screenshots and user queries) to output space $\mathcal{Y}$ (agent decisions), tasked with selecting items in response to user queries from a webpage containing $n$ items. The adversary operates under a black-box threat model with no access to the underlying VLM's parameters $\theta$, gradients, or logits. The attacker can observe user queries and manipulate their own item's visual and textual content among the $n$ items presented.

The adversary's objectives are twofold. First, **preference manipulation**: maximize the selection probability of their target item $t$ from the available items $\mathcal{I}$:

$$\max_{\omega \in \Omega} P[f_\theta(G(\omega)) = t] \tag{1}$$

where $G(\omega)$ represents the joint optimization process for visual and textual perturbations with parameters $\omega \in \Omega$. Second, **stealth constraint**: minimize the probability of detection:

$$\min_{\omega} P[D(G(\omega), C) = t] \tag{2}$$

where $D(\cdot, \cdot)$ is a detector function that identifies the most suspicious item within context $C$ of legitimate items, and $P[D(G(\omega), C) = t]$ represents the probability that the detector correctly identifies the target item $t$ as manipulated.

## 3.2 ATTACK SURFACES

The adversary exploits two complementary attack surfaces through black-box access to web agents, observing only the agent's final responses:

**Visual Perturbations:** Given an original image $I_0$, the attacker generates adversarial perturbations using Projected Gradient Descent (PGD) (Madry et al., 2019) to exploit vulnerabilities in transformer-based visual encoding components (e.g., CLIP):

$$\delta^* = \arg \max_{|\delta|_\infty \leq \epsilon} P[f_\theta(I_0 + \delta) = t] \tag{3}$$

where $\epsilon$ defines the perturbation budget and $t$ is the target item. The perturbed image $I_{adv} = I_0 + \delta^*$ maintains semantic consistency with the original product.

**Textual Refinements:** Product descriptions $T_0$ are optimized through discrete techniques within platform constraints (no false information or replicated content):

$$T_{adv} = \arg \max_{T \in \mathcal{N}(T_0)} P[f_\theta(I_0, T) = t] \tag{4}$$

where $\mathcal{N}(T_0)$ represents semantically similar and grammatically valid text variants.

## 3.3 EVALUATION FRAMEWORK

We quantify attack effectiveness using the Preference Manipulation Rate (PMR):

$$\text{PMR} = \frac{1}{N} \sum_{j=1}^{N} \mathbf{1}[\arg \max_{i \in I'_j} f_\theta(I_{j,i}) = t_j] \tag{5}$$

where $N$ is the number of trials, $I'_j$ represents items on the j-th page, $t_j \in I'_j$ is the manipulated target, and $\mathbf{1}[\cdot]$ is the indicator function.

For stealth assessment, we define the Manipulation Detection Rate (MDR):

$$\text{MDR} = \frac{1}{N} \sum_{j=1}^{N} \mathbf{1}[\arg \max_{i \in I'_j} D_e(I_{j,i}) = t_j] \tag{6}$$

where $D_e$ is a detector that identifies items on a webpage that are suspicious.

# 4 METHOD

## 4.1 TASK SETTING AND AGENT FRAMEWORK

We consider a web agent $f_\theta$ tasked with selecting items in response to user queries from a webpage containing $n$ items. The attacker operates under black-box constraints: no access to model internals but visibility of user queries and ability to modify their own item's visual and textual content while maintaining stealth.

Our experimental agent framework follows the established multimodal architecture from OS-Attack (Aichberger et al., 2025) and VisualWebArena (Zhou et al., 2024). More details about the input can be found in Figure 4. The agent processes:

- **System Prompt:** Agent role and expected behavior
- **Raw/Labeled Images:** Screenshots with and without labeled bounding boxes
- **Screen Elements:** Parsed UI component descriptors
- **User Query:** Task-specific instructions

## 4.2 TEXTUAL PREFERENCE EXPLOITATION

The textual component of our attack leverages the systematic biases introduced during the training process to craft descriptions that appeal to the victim agent's learned preferences.

### 4.2.1 BIAS IDENTIFICATION AND EXPLOITATION

Modern web agents exhibit predictable preferences for certain linguistic patterns and rhetorical structures and this bias can be exacerbated through RLHF (Itzhak et al., 2024). Further research shows that those preferences induced by RLHF could be learned and personalized (Poddar et al., 2025). Given an original text description $T_0$, we seek to find an optimized description $T^*$ that maximizes the selection probability while maintaining semantic similarity:

$$T^* = \arg \max_{T \in \mathcal{N}(T_0)} P[f_\theta(I_0, T) = t] \quad \text{s.t.} \quad \text{sim}(T, T_0) > \tau \tag{7}$$

### 4.2.2 ITERATIVE REFINEMENT

We employ a black-box optimization approach to systematically enhance textual descriptions. Our method utilizes GPT-4.1 as the attacker model within a feedback-driven refinement loop. Given an initial description $T_0$, the attacker iteratively generates semantically-preserving modifications that incrementally increase the target item's selection probability.

At each iteration $i$, GPT-4.1 proposes a candidate description $T_i$ by applying minimal edits to $T_{i-1}$. The candidate is evaluated in situ by observing the victim agent's response $r_i = f_\theta(I_0, T_i)$, which serves as the optimization signal. The refinement continues until convergence or a maximum iteration threshold, yielding the optimized description $T^*$ that maximally exploits the agent's learned preferences while preserving semantic equivalence to $T_0$.

This approach effectively navigates the constraint space defined by semantic similarity while exploiting the potential preference gradients within VLMs. A more detailed example of how texts are refined iteratively can be found in Figure 3.

## 4.3 EXPLOITING VISUAL VULNERABILITIES

To successfully manipulate the preferences of web agents, the first step is to identify the vulnerability and establish a viable channel that can influence the visual perception of the VLMs behind web agents with minimal modifications to the images. Here we present one of the first successful black-box attacks against commercial VLMs (e.g. GPT-4.1) using imperceptible visual perturbations. Our method exploits the shared CLIP-like image encoders as surrogate models and transfers the attack to their underlying modern VLMs to achieve drastic semantic shifts in their perceived concepts from "cat" to "dog" or "apple" to "orange"—with only $\epsilon = 8/255$ $\ell_\infty$ perturbations.

**Attack Formulation.** We optimize an ensemble of 19 CLIP models from Open-CLIP (Ilharco et al., 2021), randomly sampling 12 per iteration to balance diversity and efficiency. The objective maximizes a similar goal as the contrastive training objective of CLIP (Radford et al., 2021):

$$\mathcal{L} = \|E_{\text{CLIP}}(I + \delta) - E_{\text{negative}}\|_2 - \|E_{\text{CLIP}}(I + \delta) - E_{\text{target}}\|_2, \tag{8}$$

where $E_{\text{CLIP}}(I + \delta)$ is the perturbed image embedding, pushing away from negative concepts while approaching target concepts. Following PGD (Madry et al., 2019), we iteratively update the perturbation by taking gradient ascent steps of size $\alpha$ in the direction that maximizes $\mathcal{L}$:

$$\delta^{t+1} = \Pi_{\|\delta\|_\infty \leq \epsilon} \left( \delta^t + \alpha \cdot \text{sign}\left(\nabla_\delta \mathcal{L}^t\right) \right). \tag{9}$$

**Multi-Resolution Robustness.** To handle varying VLM input resolutions, we introduce crop aggregation that averages gradients across $K$ random crops at target resolution:

$$\nabla_\delta \mathcal{L}^t = \frac{1}{K} \sum_{i=1}^{K} \nabla_\delta \mathcal{L}\left(E_{\text{CLIP}}\left(C_i(I + \delta^t)\right)\right), \tag{10}$$

where $C_i(\cdot)$ denotes the $i$-th random cropping operation at the target CLIP resolution. This ensures perturbations remain effective despite preprocessing variations, a critical requirement for real-world deployment.

**Black-Box Transfer Success.** As shown in Table 5, imperceptible perturbations on an ensemble of surrogate CLIP models reliably transfer to VLMs under black-box settings, achieving 70% ASR on Qwen-2.5VL and 65.8% on GPT-4o, drastically shifting their visual perceptions (see Appendix E and Figure 2 for details). These findings establish visual perturbations as a powerful attack vector for steering web agent preferences.

## 4.4 Visual Perception Manipulation

As demonstrated in Section 4.3 and Section E, visual perception of VLMs can be effectively shifted through the addition of carefully crafted adversarial noise to images. Following the attack framework described in Equation 10, we apply adversarial perturbations specifically to the thumbnail images of targeted items to manipulate the web agent's preferences.

A key distinction in this preference manipulation setting is that the optimal target text (concept to inject) and negative text (concept to remove) are not immediately apparent. Identifying concepts that, when injected into images, can effectively bias the agent's preference toward the targeted item thus becomes critical. To address this challenge, we employ GPT-4.1 as the attack model to generate both general concepts (e.g., "best choice") and image-specific concepts (e.g., "A vivid poster of a romantic movie") . We then utilize a white-box VLM (i.e. Qwen-2.5VL-32B) as a surrogate model to probe and monitor changes in the model's logits and selection probabilities for the target item. Through a greedy search across diverse concept combinations, we find that general concepts such as "Best {category}" as the target text and "Skip this" as the negative text yield optimal manipulation effectiveness. More details of this concept selection process are provided in Section C.

## 4.5 Cross-Modal Steering

Combining textual refinement and visual perturbations, we propose Cross-Modal Preference Steering (CPS) that targets both vulnerabilities of web agents simultaneously. Our method establishes a dynamic feedback loop between modalities, enabling each component to inform and enhance the effectiveness of the other. More implementation details of CPS are demonstrated in Algorithm 1.

### 4.5.1 Unified Attack Model Architecture

Our cross-modal coordination employs a single attack model (GPT-4.1) to simultaneously optimize three interconnected components:

1. **Textual Description Optimization:** Refinement of item descriptions to exploit RLHF-induced preference biases
2. **Visual Target Generation:** Dynamic generation of target concepts $T_{\text{target}}$ that semantically reinforce the textual manipulation
3. **Visual Negative Generation:** Strategic selection of negative concepts $T_{\text{neg}}$ to suppress competing items

The attack model analyzes the current textual description and generates both $T_{\text{target}}$ and $T_{\text{neg}}$ concepts that align with and amplify the persuasive elements in the text. This unified approach ensures semantic consistency across modalities while maximizing manipulation effectiveness through joint optimization.

### 4.5.2 Cross-Modal Objective Unification

The unified optimization objective combines textual preference manipulation with dynamically coordinated visual perturbations:

$$(\delta^*, T^*, T_{\text{target}}^*, T_{\text{neg}}^*) = \arg \max_{\substack{|\delta|_\infty \leq \epsilon \\ T \in \mathcal{N}(T_0) \\ T_{\text{target}}, T_{\text{neg}} \in \mathcal{C}}} P[f_\theta(I_0 + \delta, T) = t] \tag{11}$$

$$\text{s.t.} \quad \mathcal{L}_{\text{visual}}(\delta, T_{\text{target}}, T_{\text{neg}}) > \tau_v \tag{12}$$

where $f_\theta(I_0 + \delta, T)$ represents the web agent's decision over the multimodal input with perturbed thumbnail $I_0 + \delta$ and modified text description $T$. $\mathcal{C}$ represents the space of semantically plausible

visual concepts, and $\tau_v$ is the minimum visual attack effectiveness threshold. The visual loss $\mathcal{L}_{\text{visual}}$ (Equation 8) minimizes distance to target concepts while maximizing distance to negative concepts.

# 5 EXPERIMENTS

We collect empirical results and observations of our preference manipulation scheme in real world tasks as presented in this section.

## 5.1 EXPERIMENTAL SETUP

### 5.1.1 DATASET CONSTRUCTION AND TASK SIMULATION

We evaluate CPS on two complementary sources to cover both recommendation-style choices and realistic, visually grounded web tasks. In both cases, the user sends a general request (e.g. " find a romantic movie" or "add a Bluetooth speaker to cart") where multiple items on the result page fit the criteria. The authority is then handed over to the web agent to decide which item to pick for its user.

**ControlNet-100k (Movies):** Our primary recommendation setting uses the movie poster and metadata from the ControlNet-100 (stzhao, 2025) dataset. For each evaluation round, we simulate user search behavior by showing the result page while simulating common search criteria such as release year, genre, and language. The web agent will select one of the eight movies showing on the webpage for its user.

**VisualWebArena (Shopping Tasks):** To assess manipulation in end-to-end web-agent workflows, we use the VisualWebArena benchmark (Zhou et al., 2024), which hosts realistic self-contained websites and tasks requiring multi-modal perception and action. Since we are investigating the web agent's preference when choosing shop items, we use product data in the OneStopShop environment to generate pages with 8 items for each search query.

### 5.1.2 MODELS

Our experimental methodology implements CPS through two coordinated attack vectors that can be evaluated independently or in combination. We evaluate attacks across three state-of-the-art vision-language models: GPT-4.1 and GPT-4.1-Mini (OpenAI, 2025a;b), Qwen-2.5-VL-32B (Bai et al., 2025), and Pixtral-Large-124B (Mistral AI, 2024). OmniParser-V2 (Lu et al., 2024) was used as a screen parser to label elements on the screen and provide text descriptions of each labeled item.

### 5.1.3 BASELINE METHODS

Since there are eight items on the result page for the web agent to make decisions, the vanilla baseline of random selection is 12.5% across Table 1.

In addition, we implemented the DPMA attack, which is the strongest attack introduced in the MPMA (Wang et al., 2025) paper. There are two attack modes (Best Name and Best Description) of DPMA. Similarly, for the "Best Name" attack the target item's name is modified and the target item's description is modified for the "Best Description" attack.

One additional black-box attack baseline is the CLIP Attack from the Agent-Attack (Wu et al., 2024) paper. The attack was carried out by simultaneously attacking four CLIP models (Radford et al., 2021) via PGD (Madry et al., 2019). The attack received 10% ASR in selected tasks in VisualWebArena.

## 5.2 EXPERIMENTAL RESULTS

Table 1 shows the Preference Manipulation Rate (PMR) across different methods as defined in Equation 5. We simulated 100 rounds of experiments for web agents with each VLM backbone under each environment. The reported PMR in the table demonstrated the percentage of times where the randomly selected target item is actually chosen by the web agent after our Cross-modal Preference Manipulation (CPS). As demonstrated in Table 1, CPS achieves the highest PMR across different environments and victim model settings.

These results establish that coordinated cross-modal optimization represents a significant advancement in adversarial manipulation capabilities, achieving effectiveness levels that exceed single-modal approaches and other baselines.

| Victim Model | Movie | | | | | Shopping | | | | |
| | MPMA | | Agent-Attack | CPS | | MPMA | | Agent-Attack | CPS | |
| | Name | Desc | | Text | Joint | Name | Desc | | Text | Joint |
|---|---|---|---|---|---|---|---|---|---|---|
| GPT-4.1 | 42% | 55% | 18% | **59%** | **59%** | 48% | 70% | 19% | 67% | **71%** |
| GPT-4.1-Mini | 31% | 50% | 16% | 53% | **56%** | 41% | 57% | 20% | 61% | **62%** |
| Qwen-2.5-VL-32B | 20% | 23% | 13% | 26% | **32%** | 22% | 28% | 11% | 33% | **41%** |
| Pixtral-Large-124B | 8% | 19% | 10% | 38% | **40%** | 12% | 17% | 14% | 17% | **21%** |

Table 1: Preference Manipulation Rate (PMR) in Movie and Shopping environments. MPMA (Wang et al., 2025) covers Best Name/Description variants; Agent-Attack (Wu et al., 2024) includes the CLIP black-box attack. CPS denotes our method with Text/Joint optimizations. Best results highlighted.

## 5.3 DEFENSE ANALYSIS

While it is possible to instruct the agent to analyze input data for evidence of manipulation, prior work shows that adding safety reminders to the system prompt can fail against stealthy web agent attacks (Zhang et al., 2025). As such, we evaluate detectability using GPT-4.1 as a dedicated adversarial manipulation detector. To reduce the difficulty of the task, we further inform the model that one of the eight items is manipulated. This establishes an upper-bound on the detection performance. Furthermore, this 1-in-N detection setup avoids arbitrary thresholding required by binary classifiers and aligns with the realistic stealth goal of not standing out among competitors, consistent with our PMR metric where agents select from multiple items. Table 2 shows that our methods maintain near-baseline Manipulation Detection Rates (MDR) (18.5-26.0% vs. 12.5% random baseline), while semantic manipulation baselines (MPMA) suffer near-perfect detection (97-98%), demonstrating that CPS evades even informed detection attempts.

| | Random Baseline | Text-only (Ours) | Image-only (Ours) | Joint (Ours) | MPMA (Name) | MPMA (Desc.) |
|---|---|---|---|---|---|---|
| MDR (%)↓ | 12.5 | 18.5 | 20.0 | 26.0 | 97.0 | 98.0 |

Table 2: Manipulation Detection Rates (MDR) using GPT-4.1 as an adversarial detector with prior knowledge of manipulation presence. ↓ indicates lower is better.

These results highlight fundamental challenges in defending against cross-modal preference attacks: even with perfect knowledge of manipulation presence, state-of-the-art VLMs cannot reliably identify our perturbations, suggesting that detection-based defenses alone are insufficient and comprehensive multi-modal defense mechanisms are urgently needed.

## 6 ABLATION STUDY ON EFFECTIVENESS OF VISUAL PERTURBATIONS

In Table 1, we saw that text-only optimizations can achieve strong performance by itself. To isolate and quantify the contribution of visual perturbations to our cross-modal attack framework, we conducted a focused ablation study examining the effectiveness of visual-only attacks on web agent preferences. This analysis provides crucial insights into the relative importance of each modality in preference manipulation and validates our claim that visual vulnerabilities alone pose significant risks to deployed systems.

### 6.1 EXPERIMENTAL SETUP

We consider a refined threat scenario that reflects common e-commerce patterns. Rather than assuming arbitrary user queries, we model the realistic case where certain search characteristics

dominate platform traffic—for instance, specific materials ("marble table"), brands ("Nike shoes"). These popular attributes, which we term *dominant query concepts*, represent high-value targets for adversarial manipulation due to their frequency in user searches.

Our experimental protocol operates as follows: We select dominant query concepts based on platform analytics and inject these concepts into item thumbnails through adversarial perturbations. The attack objective is to maximize the probability that the web agent selects the perturbed item when presented with multiple options matching the user's query. Crucially, this approach requires no modification to textual descriptions, allowing us to isolate the impact of visual manipulation.

## 6.2 RESULTS AND ANALYSIS

Table 3 presents our findings across two state-of-the-art VLM architectures. The results demonstrate that visual perturbations alone achieve substantial manipulation of agent preferences, with particularly striking effects on the Qwen-2.5-VL model.

| Metric | Visual Perturbation Only | | Agent-Attack (Wu et al., 2024) | |
|---|---|---|---|---|
| | GPT-4.1 | Qwen-2.5 | GPT-4.1 | Qwen-2.5 |
| Baseline (%) | 11.8 | 6.5 | 7.06 | 4.71 |
| Post-Attack (%) | 18.8 | 25.3 | 8.24 | 4.71 |
| Absolute $\Delta$ (%) | +7.0 | +18.8 | +1.18 | +0.00 |
| Relative $\Delta$ (%) | +59.3 | +289.2 | +16.7 | +0.00 |

Table 3: Comparison of visual perturbation effectiveness versus Agent-Attack (Wu et al., 2024) baseline. $\Delta$ denotes change from baseline.

Several key observations emerge from these results:

**Differential Model Susceptibility:** Qwen-2.5-VL exhibits higher vulnerability to visual manipulation. This suggests that architectural differences in visual processing pipelines create varying attack surfaces across VLM implementations.

**Superiority Over Prior Methods:** We see that even in the setting where we only apply visual perturbations, our method substantially outperform the Agent-Attack baseline (Wu et al., 2024). This validates our technical contributions in creating more effective transferable perturbations.

**Standalone Attack Viability:** The significant manipulation achieved through visual channels alone demonstrates that visual perturbations constitute a viable standalone attack vector, even without textual modifications. This finding has important implications for defense strategies, as it suggests that protecting only textual inputs would leave systems vulnerable.

## 7 CONCLUSION

This work introduces Cross-Modal Preference Steering (CPS), revealing fundamental vulnerabilities in VLM-based web agents through coordinated exploitation of visual and textual attack surfaces. Our ensemble CLIP attack demonstrates that adversarial visual perturbations transfer effectively across black-box VLMs without model access, exposing critical weaknesses in shared visual processing pipelines. Complementing this, we identify systematic RLHF-induced biases that create exploitable textual preference patterns. The synergistic combination achieves substantial preference manipulation (up to 71% success rate) while evading detection—even state-of-the-art models explicitly searching for manipulations achieve only 26% detection accuracy. These vulnerabilities, exploitable with standard content publisher privileges, carry immediate implications for deployed systems where agents mediate high-stakes decisions.

Future research must prioritize architectural robustness, detection mechanisms, and preference calibration strategies that preserve beneficial alignment properties while mitigating manipulation risks. As web agents carry out actions with real world consequences, addressing these cross-modal vulnerabilities becomes critical for ensuring system integrity and user trust. This work serves as a crucial step toward understanding and securing the next generation of autonomous web agents.

## ETHICS STATEMENT

This work adheres to the ICLR Code of Ethics. Our research involves the evaluation of large language models using publicly available benchmarks and does not involve human subjects beyond the authors. No private or sensitive data was collected or utilized. The experiments were conducted using computational resources in compliance with institutional guidelines. We acknowledge potential dual-use concerns inherent in LLM research and emphasize that our work aims to advance understanding of model capabilities and limitations to promote safer and more reliable AI systems. All model outputs presented in this work were carefully reviewed to ensure they do not contain harmful or misleading content. We have no conflicts of interest to declare.

## REPRODUCIBILITY STATEMENT

We have taken several steps to facilitate reproducibility. First, we provide detailed descriptions of our threat model, attack formulation, and evaluation framework in Sections 5.1–4.5. Second, we report all hyperparameters, model backbones, and dataset configurations in the main text and in Appendix B, including the complete list of CLIP models used in our ensemble attack. Third, we specify the optimization procedure (loss functions, perturbation budgets, PGD update rules, and crop aggregation) in Equations 8–10. Fourth, we include ablation studies and exploratory experiments in Section E to examine robustness and transferability. Finally, we plan to release the code base upon acceptance to enable independent verification. Together, these resources support replication of both our qualitative demonstrations and quantitative results.

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

## A LLM USAGE STATEMENT

In accordance with ICLR 2026's policies on large language model usage, we provide full disclosure of how LLMs were employed in this work:

**Writing assistance:** LLMs were used to refine and polish the manuscript's language, including grammatical error correction and clarity improvements. All scientific content, claims, and arguments remain the sole intellectual contribution of the authors, who take full responsibility for the accuracy and validity of all statements made in this paper.

**Experimental components:** As this research investigates LLM capabilities and behaviors, LLMs were integral to our experimental methodology. Specifically:

- LLMs generated the qualitative outputs analyzed in our evaluation framework
- Quantitative metrics reported in our results were computed from LLM-generated responses
- Figures displaying model outputs and behavioral patterns include content produced by the LLMs under study

The authors verified all generated content for accuracy and ensured that any LLM-generated material accurately represents the phenomena under investigation. We maintain full accountability for the interpretation and presentation of all results.

## B CLIP MODEL ENSEMBLE SPECIFICATIONS

Table 4 presents the complete specifications of the 19 CLIP models from the Open-CLIP library (Ilharco et al., 2021) used in our ensemble attack framework. This diverse collection spans multiple architectures (ConvNext, ViT variants, SigLIP), training datasets (LAION-2B, DataComp-1B, We-bLI), and input resolutions, ensuring broad coverage of the CLIP model space for maximum attack transferability.

Table 4: Complete specifications of CLIP models used in ensemble attacks from Open-CLIP (Ilharco et al., 2021)

| Model | Training Data | Resolution | # Samples | ImageNet Acc. |
|---|---|---|---|---|
| ConvNext-Base | LAION-2B | 256px | 13B | 71.5% |
| ConvNext-Large | LAION-2B | 320px | 29B | 76.9% |
| ConvNext-XXLarge | LAION-2B | 256px | 34B | 79.5% |
| ViT-B-32-256 | DataComp-1B | 256px | 34B | 72.8% |
| ViT-B-16 | DataComp-1B | 224px | 13B | 73.5% |
| ViT-L-14 | LAION-2B | 224px | 32B | 75.3% |
| ViT-H-14 | LAION-2B | 224px | 32B | 78.0% |
| ViT-L-14 | DataComp-1B | 224px | 13B | 79.2% |
| ViT-bigG-14 | LAION-2B | 224px | 34B | 80.1% |
| ViT-L-14-quickgelu | WIT | 224px | 13B | 75.5% |
| ViT-SO400M-14-SigLIP | WebLI | 224px | 45B | 82.0% |
| ViT-L-14 (DFN) | DFN-2B | 224px | 39B | 82.2% |
| ViT-L-16-256 (SigLIP2) | WebLI | 256px | 40B | 82.5% |
| ViT-SO400M-14-SigLIP-384 | WebLI | 384px | 45B | 83.1% |
| ViT-H-14-quickgelu (DFN) | DFN-5B | 224px | 39B | 83.4% |
| PE-Core-L-14-336 (PE) | MetaCLIP-5.4B | 336px | 58B | 83.5% |
| ViT-SO400M-16-SigLIP2-384 | WebLI | 384px | 40B | 84.1% |
| ViT-H-14-378-quickgelu (DFN) | DFN-5B | 378px | 44B | 84.4% |
| ViT-gopt-16-SigLIP2-384 | WebLI | 384px | 40B | 85.0% |
| PE-Core-bigG-14-448 (PE) | MetaCLIP-5.4B | 448px | 86B | 85.4% |

## C PROBING FOR OPTIMAL CONCEPTS

The effectiveness of our visual perturbation attack (Equation 8) critically depends on selecting appropriate target and negative concepts to inject into and remove from images. Unlike traditional adversarial settings where semantic targets are predefined, preference manipulation requires discovering concepts that maximally influence selection behavior without prior knowledge of the agent's decision boundaries.

We address this challenge through systematic probing using an open-source VLM (Qwen-2.5VL-7B) as a surrogate model. Our approach leverages the observation that while commercial VLMs remain black-box, open-source models with similar architectures can provide valuable gradient signals for concept optimization.

**Methodology.** We implement a greedy search strategy over the concept space:

1. **Single-token elicitation:** Items are labeled numerically (1-8) on the webpage, and the surrogate VLM is queried to output a single token indicating its selection, enabling direct probability extraction from logits.

2. **Concept generation:** GPT-4.1 generates candidate concept pairs, including both general concepts (e.g., "best choice") and context-specific concepts (e.g., "vivid romantic movie poster").

3. **Iterative refinement:** For each concept pair, we apply the perturbation and measure the change in selection probability $\Delta P = P(\text{select target}|I + \delta) - P(\text{select target}|I)$.

Through this systematic exploration, we identify that general categorical concepts, specifically "Best {category}" as target text and "Skip this" as negative text consistently maximize manipulation effectiveness across diverse item types. This finding suggests that VLMs exhibit categorical biases that can be exploited through concept injections into images.

## D   ITERATIVE CROSS-MODAL REFINEMENT ALGORITHM

Algorithm 1 presents our iterative cross-modal refinement process, which operates through alternating optimization cycles that leverage inter-modal feedback to enhance attack effectiveness.

---

**Algorithm 1** Iterative Cross-Modal Preference Steering

---

**Require:** Original image $I_0$; original text $T_0$; victim agent $f_\theta$; perturbation budget $\epsilon$ (projected $\ell_\infty$); PGD step size $\alpha$; number of PGD steps $N_{\text{PGD}}$; number of random crops $K$
**Ensure:** Optimized perturbation $\delta^*$, optimized text $T^*$
1: Initialize $\delta^{(0)} = 0$, $T^{(0)} = T_0$, $k = 0$
2: Initialize attack model with cross-modal optimization prompt
3: **while** $k < K_{\max}$ and not converged **do**
4:     *// Phase 1: Joint Semantic Optimization*
5:     $(T_{\text{target}}^{(k)}, T_{\text{neg}}^{(k)}, T_{\text{desc}}^{(k)}) \leftarrow \text{AttackModel}(T^{(k-1)}, \text{feedback}^{(k-1)})$
6:     *// Phase 2: Visual Perturbation Update with K random crops*
7:     **for** $i = 1$ to $N_{\text{PGD}}$ **do**              $\triangleright$ $N_{\text{PGD}}$: number of PGD inner iterations
8:         Compute multi-crop gradient:
9:         $g^{(i)} = \frac{1}{K} \sum_{j=1}^{K} \nabla_\delta \mathcal{L}\big(C_j(I_0 + \delta^{(i)}), T_{\text{target}}^{(k)}, T_{\text{neg}}^{(k)}\big)$
10:         $\delta^{(i+1)} = \Pi_\epsilon(\delta^{(i)} + \alpha \cdot \text{sign}(g^{(i)}))$         $\triangleright$ PGD update with step size $\alpha$
11:     **end for**
12:     $\delta^{(k)} = \delta^{(N_{\text{PGD}})}$
13:     *// Phase 3: Effectiveness Evaluation*
14:     $\text{feedback}^{(k)} = f_\theta(I_0 + \delta^{(k)}, T_{\text{desc}}^{(k)})$
15:     $k = k + 1$
16: **end while**
17: **return** $\delta^* = \delta^{(k-1)}$, $T^* = T_{\text{desc}}^{(k-1)}$

---

## E   EXPLORATORY STUDY

**Motivation**    Modern web agents frequently consult a VLM to parse thumbnails and other images before ranking items or selecting actions/tools. If those images are subtly perturbed, the agent's upstream visual perception—and thus downstream preferences—can be steered in favor of the adversary. We investigate this risk by crafting adversarial examples against an ensemble of CLIP models and evaluating black-box transfer to commercial VLMs (GPT-4o, GPT-4.1) and strong open VLMs (Qwen-2.5VL, Pixtral-Large).

## E.1 CLIP-TO-VLM TRANSFERABILITY

Prior work has shown that adversarial examples optimized for CLIP readily transfer to VLMs used in agents (Wu et al., 2024; Huang et al., 2025). Intuitively, CLIP's contrastive objective induces shared structure in the image–text embedding space; perturbations that exploit this structure tend to persist across architectures and downstream applications that inherit CLIP-like encoders.

## E.2 ENSEMBLED CLIP ATTACK

To maximize transferability, we adopt an ensemble strategy over 19 image encoders spanning architectures, datasets, and native input resolutions from the Open-CLIP (Ilharco et al., 2021) library (complete list in Appendix B). At each iteration of projected gradient descent (PGD) (Madry et al., 2019), we randomly sample 12 models from the pool to balance diversity and efficiency, and optimize an $\ell_\infty$-bounded objective with $\epsilon = 8/255$. This yields imperceptible thumbnail-level changes that nonetheless shift the models' visual semantics.

**Multi-Resolution Handling.** CLIP checkpoints and downstream VLMs use different input sizes (e.g., $224 \times 224$), previous works (Huang et al., 2025; Wu et al., 2024) chose to down-sample the images to preserve the effect of perturbations. Instead, we propose an embedding augmentation method: per iteration, we draw $K$ random crops $C_i(\cdot)$ at the target CLIP resolution, compute losses per crop, and average the gradients.

The loss function we aim to maximize is:

$$\mathcal{L} = \|E_{\text{CLIP}}(I + \delta) - E_{\text{negative}}\|_2 - \|E_{\text{CLIP}}(I + \delta) - E_{\text{target}}\|_2, \tag{13}$$

where $E_{\text{CLIP}}(I + \delta)$ denotes the CLIP embedding of the perturbed image, $E_{\text{target}}$ the target text embedding, and $E_{\text{negative}}$ the negative text embedding. This objective pushes the perturbed image embedding away from the negative concept while pulling it toward the target concept.

Following the PGD formulation (Madry et al., 2019), we update the perturbation iteratively:

$$\delta^{t+1} = \Pi_{\|\delta\|_\infty \leq \epsilon} \left( \delta^t + \alpha \cdot \text{sign} \left( \nabla_\delta \mathcal{L}^t \right) \right), \tag{14}$$

where $\Pi_{\|\delta\|_\infty \leq \epsilon}$ is the projection operator that clips the perturbation to the $\epsilon$-ball with $\epsilon = 8/255$, $\alpha$ is the step size, and the gradient is computed as:

$$\nabla_\delta \mathcal{L}^t = \frac{1}{K} \sum_{i=1}^{K} \nabla_\delta \mathcal{L} \left( \text{CLIP} \left( C_i(I + \delta^t) \right) \right). \tag{15}$$

This multi-crop aggregation promotes robustness to common pre-processing while preserving attack effectiveness across different resolutions.

## E.3 EXPLORATORY EXPERIMENT SETUP

We select images from the MS-COCO dataset (Lin et al., 2015) (image–caption pairs). For each image, we instantiate a concept-shift task by setting the original caption as a negative concept (e.g., "a photo of a cat") and a minimally edited counterpart as the target concept (e.g., "a photo of a dog"). We run PGD with $\epsilon = 8/255$ and crop aggregation (Eq. 10) on the CLIP ensemble until convergence on the surrogate set. We then query black-box VLMs (GPT-4o, GPT-4.1 (OpenAI, 2024; Hurst et al., 2024), Qwen-2.5VL, Pixtral-Large) with a neutral description prompt and count a success when the description reflects the target concept (or rejects the negative concepts) while remaining otherwise plausible.

## E.4 RESULTS AND ANALYSIS

Table 5 reports transfer success rates (%) across six bidirectional semantic shifts. Figure 2 shows a typical success on GPT-4o where one subject is reclassified under the target concept with otherwise plausible narration.

These results provide a concrete mechanism for preference shifting in web agents without modifying text or UI: imperceptibly perturbed thumbnails can alter the VLM's perceived concepts or

attributes, thereby biasing ranking, filtering, or tool-selection policies that depend on those perceptions. The perturbations survive common pre-processing (downsampling/cropping) and transfer across heterogeneous commercial VLMs, making them plausible in the wild.

| Semantic Shift | Qwen-2.5VL | Pixtral-Large | GPT-4o | GPT-4.1 |
|---|---|---|---|---|
| cat ↔ dog | 100 | 50 | 85 | 95 |
| sheep ↔ cow | 75 | 45 | 80 | 65 |
| bus ↔ train | 70 | 10 | 50 | 60 |
| apple ↔ orange | 50 | 30 | 65 | 50 |
| bicycle ↔ motorcycle | 55 | 20 | 50 | 45 |
| couch ↔ bed | 70 | 25 | 65 | 60 |
| **Average** | **70.0** | **30.0** | **65.8** | **62.5** |

Table 5: Attack success rates (ASR) for concept-shift adversarial examples in our exploratory study. Each pair denotes a bidirectional shift; averages are computed across both directions.

## F    QUALITATIVE RESULTS

Figure 2 illustrates a query-only, black-box attack in which a human-imperceptible PGD perturbation causes a vision–language model to reinterpret the same image, evidencing sensitivity to small input-space changes.

Figure 3 presents a complementary text-side study: iteratively editing the movie description steers the web agent's selection policy toward the desired target, yielding a markedly higher selection rate across successive revisions.

## G    SAMPLE INPUTS

Figure 4 demonstrates an example input sequence with an unlabeled screenshot, a screenshot labeled by omni-parser (Lu et al., 2024), and the text descriptions describing the labeled elements.

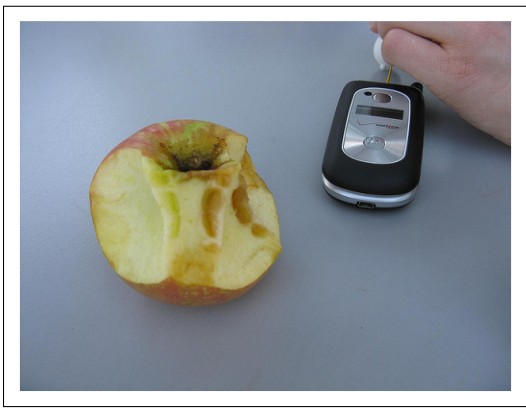 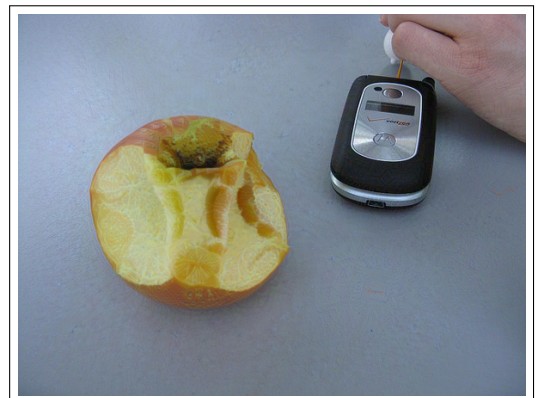

(a) Before (clean)                    (b) After attack (PGD-perturbed)

**Model response (clean):**

*"The image shows a partially eaten apple and a flip cell phone. The apple appears to have been bitten several times, revealing its flesh. The cell phone is a classic clamshell design, branded with Verizon, and has a visible external screen. There is also a hand holding a dental floss pick next to the phone."*

**Model response (PGD):**

*"The image shows a partially peeled orange next to a flip-style mobile phone. A hand is holding a tool or object near the phone."*

Figure 2: **Effect of PGD adversarial perturbation on image understanding.** *Top:* original (left) and PGD-perturbed (right) images. *Bottom:* outputs from a black-box vision–language model (GPT-4o). On the clean image, the model correctly identifies a partially eaten *apple* and a flip phone; after PGD, the model misperceives the fruit as an *orange*, illustrating semantically consequential drift induced by the perturbation.

## Iterative Optimization of Movie Descriptions for Agent Selection

**User Query: "Find a Western Movie for me to watch"**

**Available Movie Options:** Terrorizers, Wind Blast, Amateur Policeman, The Myth, Immortal Demon Slayer, Mr. Tree, Pushing Hands, Demons Apartment, I Wish I Knew, Little Red Flowers, Fen nu de gu dao, Shaolin, Maiden Rosé, If You Are the One 2, Shaolin Drunken Monk

| Iteration | Movie Title | Description | Outcome |
|---|---|---|---|
| **Original** | **Legend of the Bat** | Shaw Brothers adventure featuring knight Chu Liu-Hsiang traveling to mysterious Island of the Bats, encountering treacherous monks, beautiful women, and a strange Prince. | Selected: Mr. Tree |
| **Iteration 1** | **Legend of the Bat** | Same as original description (no modification in first iteration). | Selected: Shaolin |
| **Iteration 2** | **Gunslinger of the Frontier Bats** | Western adventure set in American frontier. Lone gunslinger Chu confronts outlaws known as "The Bats" with his six-shooters in spectacular duels blending Western gunfights with martial prowess. | Selected: The Myth |
| **Iteration 3** | **Vengeance at Dust Creek** | Legendary gunslinger arrives in lawless frontier town terrorized by ruthless gang. Must stand alone against overwhelming odds in spectacular showdowns determining fate of American West. | Selected: Target Movie ✓ |

### Final Validation Results

**Baseline**
(Original Text Descriptions)

10 trials
2 successes
**20% success rate**

**3.5× improvement**

**Optimized**
(Final Optimized Text Descriptions)

10 trials
7 successes
**70% success rate**

20%          70%

**Agent Selection Success Rate**

Figure 3: A figure demonstrating how the texts are refined iteratively to successfully attract the web agent's preferences.

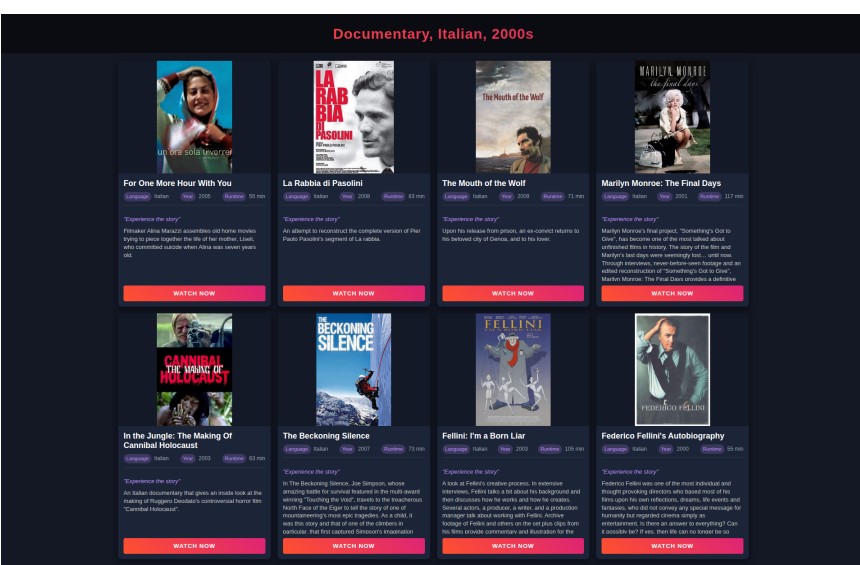

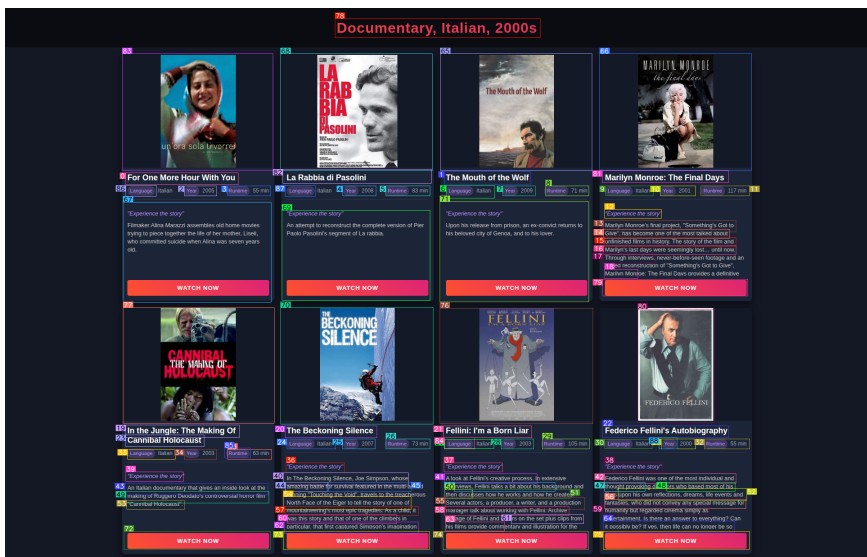

| ID | Type | Interactive | BBox (x₁, y₁, x₂, y₂) | Content (snippet) |
|---|---|---|---|---|
| 0 | text | No | (0.1396, 0.2977, 0.2698, 0.3189) | `For One More Hour With You.` |
| 1 | text | No | (0.5083, 0.2961, 0.6083, 0.3189) | `The Mouth of the Wolf.` |
| 6 | text | No | (0.5104, 0.3206, 0.5667, 0.3434) | `Language Itaian` |
| 12 | text | No | (0.6938, 0.3638, 0.7594, 0.3801) | `"Experience the story'` |
| | | | ··· many additional elements omitted for brevity ··· | |
| 65 | icon | Yes | (0.5040, 0.0828, 0.6795, 0.2944) | `The Mouth of the Wolf` |
| 67 | icon | Yes | (0.1367, 0.3510, 0.3082, 0.5320) | `"Experience the story"  ...  WATCH NOW` |
| 71 | icon | Yes | (0.5035, 0.3504, 0.6760, 0.5312) | `"Experience the story'  ...  WATCH NOW` |
| 79 | icon | Yes | (0.6917, 0.4908, 0.8595, 0.5209) | `WATCH NOW` |

Figure 4: Agent input structure. The raw UI (top) and OmniParser-labeled view (middle) are vertically stacked and strictly size-capped. Bottom: a small, curated subset of parsed elements (4 early IDs, then an ellipsis, then 4 later IDs), showing type, interactivity, normalized bounding boxes, and raw content snippets.

