# OpenReview forum: "Cross-Modal Content Optimization for Steering Web Agent Preferences"
_ICLR.cc/2026/Conference — Submitted to ICLR 2026_

### Official Review · Reviewer_Hgoh · 2025-10-22

**Soundness:** 2
**Presentation:** 3
**Contribution:** 3
**Rating:** 4
**Confidence:** 3

**Summary:**

The paper proposes **CPS (Cross‑Modal Preference Steering)**, an attack for manipulating the choices of multimodal web agents by jointly optimizing **images** and **text** associated with a single listing (e.g., a product or movie). On the **visual** side, CPS learns small, transferable perturbations on an ensemble of CLIP‑like encoders using multi‑crop PGD with an $\ell_\infty$ budget (typically noted as $8/255$). On the **text** side, CPS iteratively edits the item description to exploit LLM‑induced biases while claiming to preserve semantics and platform policy compliance. The attacker is assumed to control only their own listing’s thumbnail(s) and text; the victim agent is treated as a black box.

Experiments use two environments (a movie‑selection page and a shopping scenario) and measure **Preference Manipulation Rate (PMR)**. CPS is reported to outperform baselines, including in **visual‑only** ablations, and to transfer from CLIP surrogates to proprietary VLMs. A small detection study (a prompted LLM detector told that exactly one item is adversarial) suggests low **Manipulation Detection Rate (MDR)**.

I find the cross‑modal framing relevant and the visual objective technically plausible. However, I’m not convinced the **practicality** and **stealth** claims are adequately supported under realistic constraints, especially for the textual component.

**Strengths:**

- **Originality / Framing.** I like the positioning of content‑steering as cross‑modal optimization (image + text) as it aligns with how real agents consume web content.
- **Technical plausibility (visual).** Multi‑crop, ensemble‑CLIP PGD is a sensible black‑box strategy to mitigate unknown preprocessing; the $\ell_\infty$ budget is explicit, and transfer to proprietary VLMs suggests generality.
- **Empirical signal.** Visual‑only ablations show images alone can tilt preferences; cross‑model transfer indicates the method isn’t tightly coupled to one backbone.
- **Clarity & scope.** The manuscript is mostly clear, figures are helpful, and the evaluation spans two environments rather than a single toy task.
- **Significance (conditional).** If claims held under realistic constraints and stronger defenses, the impact for agent‑safety and content‑moderation would be substantial.

**Weaknesses:**

1. I think the largest gap is between the paper’s **practicality narrative** and the operational reality it actually assumes. The method leans on conditions that are hard to guarantee in production: awareness of or conditioning on the user’s current goal, the ability to **probe the deployed agent repeatedly** and **adapt** based on its responses, and the freedom to **continuously modify** a listing’s text and thumbnail until the manipulation “sticks.” In real marketplaces, repeated probing runs into rate limits and caching; text/image edits are often throttled or moderated; rendering to the *same* user session isn’t instantaneous or even guaranteed; and you rarely see the end‑user’s exact prompt. Even if the visual perturbations can be prepared offline, the paper’s **textual loop** explicitly depends on multiple online interactions. I would need to see results under **tight edit/eval budgets** (e.g., one shot or a handful of cycles), without access to user‑specific prompts, and with realistic delays and platform constraints to believe this is meaningfully more “practical” than the attacks the authors criticize.

2. Closely related—and, to me, just as serious—is the **tension between the claimed “semantics‑preserving” textual edits** and the examples shown. The narrative says the text stays truthful and policy‑compliant, but the qualitative case freely drifts genre/attributes. If the attack is allowed to **persuade with unconstrained language** (e.g., “this is the best choice for you,” or by reframing content to match the agent’s latent preferences), then the hard part isn’t *making* the agent pick the item—of course persuasive text can sway an LLM agent—it’s doing so while **remaining faithful to the original item** and staying within platform rules. That’s the standard they set for themselves, and I don’t see it met. Conversely, if the authors lean into the persuasion framing, then **detectability becomes the core question**: show that such edits aren’t trivially caught by policy filters, cross‑modal consistency checks, or simple rule‑based moderation. Right now the paper sits in an uncomfortable middle: enough freedom to make the attack work, but not enough constraint to substantiate the “benign/stealthy” claim.

3. I also find the **stealth/detection evaluation too narrow** to support strong claims. A single large model, prompted as a detector and told in advance that exactly one of the items is adversarial, is not a robust proxy for a platform defense. That prior (“1‑in‑$K$”) makes the task much easier than **binary detection** in the wild, and using a model from the same family as the agent invites correlated blind spots. If stealth is part of the headline, I would expect heterogeneous detectors (a vision‑only forensics model, an independent VLM, a cross‑modal consistency scorer), *plus* human raters for perceptual visibility, and reporting that looks like **AUC/ROCs and calibrated operating points**, not just a single MDR number.

4. On **reproducibility and statistical rigor**, the paper comes up short. The visual attack hinges on multi‑crop PGD with an $\ell_\infty$ budget, but the **numerical hyperparameters** that actually matter (step size, number of steps, number of crops, outer‑loop iterations/early stopping) are not all specified. The text‑refinement loop similarly lacks **clear stopping rules, query counts, and search/beam parameters**. Main tables appear to report single percentages (often over 100 trials) **without uncertainty**. Also, there are no confidence intervals or variance across seeds/sessions.

5. There’s also a bit of **threat‑model drift** in the methodology. The paper self‑identifies as black‑box with respect to the victim agent, yet it relies on **surrogate models** (and, at points, surrogate introspection) to guide optimization and concept search. That’s a reasonable engineering choice, but in practice an attacker may not have access to a high‑fidelity surrogate—or, more importantly, a way to validate that the surrogate’s gradients align with the victim’s behaviors on the specific UI. The paper can benefit from squarely addressing how much of the reported success depends on this surrogate alignment, and how often the attack fails when the surrogate is misspecified.

7. The **external validity** of the evaluation is limited. The two environments—a synthetic movie page and a benchmark shopping setting—are useful, but they don’t capture the messiness of live e‑commerce or content platforms. Real agents often mix page perception with **non‑visual ranking signals** (price, stock, popularity, reputation), and small changes in UI or copy can be overridden by those signals. I don’t see evidence that CPS meaningfully moves choices when such competing signals are present, or when the page layout, fonts, and component positions deviate from the testbed.

8. On the **imperceptibility** claim for images, I remain unconvinced without **perceptual metrics** or a human study. An $\ell_\infty$ budget of $8/255$ is frequently used, but depending on content and scale it can still introduce visible speckling or edge artifacts, especially after resampling. A small LPIPS/SSIM/PSNR report (or a blinded user test) would make the “imperceptible” adjective feel earned rather than asserted.

**Questions:**

Every item raised in the Weaknesses section can be viewed as a question for the authors (I'm particularly interested in the first 2 items).
I may well be mistaken on several of these points, and I would sincerely appreciate clarification or correction wherever appropriate.
If the authors can address or resolve even part of these concerns—whether by showing that I misunderstood something or by providing additional detail—it would be very helpful.

**Details Of Ethics Concerns:**

*   I see clear dual‑use risk: the method is explicitly designed to manipulate autonomous agents on consumer platforms. If operationalized, it could be used to steer purchases or decisions without users’ awareness.
*  The attack scenario plausibly conflicts with typical marketplace ToS (manipulative content, false or misleading claims, adversarial modifications). If data, code, or demos encourage violating site policies or scraping constraints, that also warrants scrutiny.
*  The paper offers step‑by‑step guidance for cross‑modal manipulation. Without strong guardrails (e.g., release restrictions, red‑teaming, mitigation guidance), I worry about straightforward weaponization.
*  If the authors release perturbations, prompts, or a live demo, I’d like to see a concrete harm‑mitigation plan (rate‑limits, gating, detection baselines) and clarity around any human evaluations (consent/compensation).

---

> ### Author Response · Authors · 2025-11-21
> **Response to Reviewer Hgoh Part I**
>
> We thank the reviewer for the valuable feedback on improving this paper!
> Below are our responses to your questions and concerns.
>
> > I think the largest gap is between the paper’s practicality narrative and the operational reality it actually assumes. The method leans on conditions that are hard to guarantee in production: awareness of or conditioning on the user’s current goal, the ability to probe the deployed agent repeatedly and adapt based on its responses, and the freedom to continuously modify a listing’s text and thumbnail until the manipulation “sticks.” In real marketplaces, repeated probing runs into rate limits and caching; text/image edits are often throttled or moderated; rendering to the same user session isn’t instantaneous or even guaranteed; and you rarely see the end‑user’s exact prompt. Even if the visual perturbations can be prepared offline, the paper’s textual loop explicitly depends on multiple online interactions. I would need to see results under tight edit/eval budgets (e.g., one shot or a handful of cycles), without access to user‑specific prompts, and with realistic delays and platform constraints to believe this is meaningfully more “practical” than the attacks the authors criticize.
>
> > **The method leans on conditions that are hard to guarantee in production: awareness of or conditioning on the user’s current goal**
>
> The attack does not assume that the attacker can observe user queries. Instead, the attack is established on the idea that the attacker knows what kind of products they are trying to “promote”, hence it is reasonable for the attacker to successfully infer the user query that might lead to the search result pages containing the target item.
>
> As mentioned in the Introduction Section starting from line 53, the attacker assumes that the user ask for a particular type of product. (For example, “Buy A bluetooth speaker”) Then multiple items (For example, different models of “bluetooth speakers”) will surface on top of the page that all match the search criteria. Our preference manipulation takes place at this step and aims at maximizing the likelihood of the target item being chosen by the web agent.
>
> In addition, to better simulate the real-world market scenario, our evaluation of preference manipulation is done by pairing the target item with random items in the same search category for each iteration. This setting ensures the robustness and practicality of our preference manipulation in real-world environments.
>
>
> > **I would need to see results under tight edit/eval budgets (e.g., one shot or a handful of cycles)**
>
>
>
> Considering repeated probes may trigger throttling and rate limits, we have set the condition that a maximum of 20 queries are allowed to be sent to the web_agent (e.g. GPT-4.1) for optimization for each target item. On average 10.90 iterations were used per target item. The setting of limited trials is almost identical to black-box jailbreak attacks such as Persuation Attack [1], where a maximum of 10 trials is allowed for each round of textual refinement for jailbreak attacks. Given the higher alertness on jailbreak attack's harmful prompts across all platforms, it makes our setting of using 10-20 maximum trials more practical as our query have much higher evasiveness and can easily blend in with normal queries from other common users.
>
> In addition, we did limit the budget to 5 and 10 queries and here are the results (random baseline = 12.5%):
>
> | Method | Model    | Dataset | Max_iter = 5 | Max_iter = 10 | Max_iter = 20 |
> |--------|----------|---------|--------------|---------------|----------------|
> | CPS    | GPT-4.1  | Movie   | 39.8%        |     52.2%     | 59%            |
>
> The result shows that limiting the budget does have some impact on the attack effectiveness.
>
> [1] How Johnny Can Persuade LLMs to Jailbreak Them: Rethinking Persuasion to Challenge AI Safety by Humanizing LLMs https://aclanthology.org/2024.acl-long.773

---

> ### Author Response · Authors · 2025-11-21
> **Response to Reviewer Hgoh Part II**
>
> > Closely related—and, to me, just as serious—is the tension between the claimed “semantics‑preserving” textual edits and the examples shown. The narrative says the text stays truthful and policy‑compliant, but the qualitative case freely drifts genre/attributes. If the attack is allowed to persuade with unconstrained language (e.g., “this is the best choice for you,” or by reframing content to match the agent’s latent preferences), then the hard part isn’t making the agent pick the item—of course persuasive text can sway an LLM agent—it’s doing so while remaining faithful to the original item and staying within platform rules. That’s the standard they set for themselves, and I don’t see it met. Conversely, if the authors lean into the persuasion framing, then detectability becomes the core question: show that such edits aren’t trivially caught by policy filters, cross‑modal consistency checks, or simple rule‑based moderation. Right now, the paper sits in an uncomfortable middle: enough freedom to make the attack work, but not enough constraint to substantiate the “benign/stealthy” claim.
>
> The semantic-preserving textual edits follow the settings of the MPMA [2] paper, which iteratively refine text to improve both evasiveness and effectiveness in preference manipulation. Because we are one of the earliest works introducing the vulnerabilities of preference manipulation attacks, admittedly there is not enough existing work to thoroughly test for the evasiveness of our textual edits. That is also one of the reasons why we did not include "textual-edits" as one of our main contributions in the Introduction Section.
>
> Our main contribution that we would like to emphasize is that we are the first to develop a new pathway for adversarially attacking Visual-Language Models' perceptions via imperceptible perturbation under black-box circumstances that are robust to different processing, including compressing and resizing. One tangible illustration is that we can:
>
>     1) Zoom in on Figure 2 of the PDF of our paper on page 17
> 	2) Take a screenshot and crop the adversarial image out for testing
> 	3) Send the cropped image to the ChatGPT online platform while asking “Describe the image”
>
> We have checked multiple times and the latest GPT-5.1 model will still mistakenly believe that the apple in the screenshot is an orange, and this highly robust black-box visual perception manipulation is one of the biggest contributions in our work. No other works have successfully done this to proprietary models before. We encourage reviewers to independently verify this claim using the reproduction steps provided as we are confident in our method's robustness.
>
> This robust adversarial attack opens up the possibility of directly injecting certain concepts/characteristics into thumbnail images via imperceptible perturbations. As demonstrated in Section 6, it is also feasible to inject certain popular characteristics (e.g. **Nike** shoe or **Marble** table) into thumbnail images of the target products to increase the possibilities of it being selected by web agents (Table 3). In addition, it is also shown to be possible to jointly optimize both the visual and textual attacks to receive an even higher rate of success (Table 1).
>
> [2] MPMA: Preference Manipulation Attack Against Model Context Protocol https://arxiv.org/pdf/2505.11154
>
> > I also find the stealth/detection evaluation too narrow to support strong claims. A single large model, prompted as a detector and told in advance that exactly one of the items is adversarial, is not a robust proxy for a platform defense. That prior (“1‑in‑
> ”) makes the task much easier than binary detection in the wild, and using a model from the same family as the agent invites correlated blind spots. If stealth is part of the headline, I would expect heterogeneous detectors (a vision‑only forensics model, an independent VLM, a cross‑modal consistency scorer), plus human raters for perceptual visibility, and reporting that looks like AUC/ROCs and calibrated operating points, not just a single MDR number.
>
> Thank you so much for your suggestions. The idea to explicitly tell the detector to find the one suspect was that a binary detector might find all items not suspicious, and the (“1‑in‑”) setting was meant to be an upper bound.
>
> Following your suggestions, we have conducted extra experiments that uses an independent VLM (i.e. GPT-4.1) as a cross‑modal consistency detector, a binary detector and a risk scorer.
>
> Here are the results on detecting target items after joint optimizations of text and image:
>
> Consistency check detection rate: 25.0%
>
> Binary detector detection rate: 21.5%
>
> In lieu of a ROC curve, we can show the top-1, top-2 and top-5 detection rates with a risk scorer:
>
> Top-1 Detection Rate: 21.5%
>
> Top-2 Detection Rate: 27.2%
>
> Top-5 Detection Rate: 54.3%
>
> If time permits, we are going to conduct the human evaluation experiments on perceptability and report in the final camera-ready version.

---

> ### Author Response · Authors · 2025-11-21
> **Response to Reviewer Hgoh Part III**
>
> > On reproducibility and statistical rigor, the paper comes up short. The visual attack hinges on multi‑crop PGD with an budget, but the numerical hyperparameters that actually matter (step size, number of steps, number of crops, outer‑loop iterations/early stopping) are not all specified. The text‑refinement loop similarly lacks clear stopping rules, query counts, and search/beam parameters. Main tables appear to report single percentages (often over 100 trials) without uncertainty. Also, there are no confidence intervals or variance across seeds/sessions.
>
> Thank you for pointing out the hyperparameters. Here are the details:
>
> Visual Perturbations (PGD):
>
> L∞ = 8/255,
> step size = 0.5/255,
> number of steps = 50,
> number of crops = 20,
> number of surrogate CLIP models = 12
>
> Textual perturbation:
>
> Maximum iteration = 20
> Stopping rule: The target item being chosen
>
> The information will also be included in the camera-ready version of this paper.
>
>
> > There’s also a bit of threat‑model drift in the methodology. The paper self‑identifies as a black‑box with respect to the victim agent, yet it relies on surrogate models (and, at points, surrogate introspection) to guide optimization and concept search. That’s a reasonable engineering choice, but in practice an attacker may not have access to a high‑fidelity surrogate—or, more importantly, a way to validate that the surrogate’s gradients align with the victim’s behaviors on the specific UI. The paper can benefit from squarely addressing how much of the reported success depends on this surrogate alignment, and how often the attack fails when the surrogate is misspecified.
>
>
> Thank you for your question!
>
> In the case of black-box visual attack, we are adopting an ensemble of CLIP encoders as our surrogate models. Such models are easily highly accessible through the open-clip library. The attack on an ensemble of CLIP models is more transferable to other unseen encoders in current commercial models because:
>
> At a high level, there are shared vulnerabilities among visual models that are trained with contrastive objectives including CLIP. For example, one paper in ICLR 2023 (https://iclr.cc/virtual/2023/poster/11277) found that it is possible to poison contrastive learning classification models such as SimCLR, MOCO, and BYOL with imperceptible adversarial perturbations via PGD as well. It means that at a high level these contrastively trained models such as CLIP might share a lot of vulnerabilities that can be easily exploited.
>
> A second and more practical explanation is that there are limited amount of captioned image training datasets such as LAION-2B. At the same time, there are also great similarities between the architectures of image encoders on different VLMs. Therefore, although different VLMs claim to have different image encoders, these encoders still share substantial commonalities in their training data and architectures. Such shared features create common vulnerabilities that can be optimized and learned via attacking multiple (e.g. 12 ) image encoders at the same time. Those two factors both help explain the high transferability of our Attack.
>
> In the case of concept probing, we only need one white-box VLM to serve as the victim once to find useful concepts to inject. The probing serves as a one-time exploratory analysis that might not be used in all attack generation steps. Therefore, the impact of this surrogate model is minimal to our overall black-box assumption.
>
> > The external validity of the evaluation is limited. The two environments—a synthetic movie page and a benchmark shopping setting—are useful, but they don’t capture the messiness of live e‑commerce or content platforms. Real agents often mix page perception with non‑visual ranking signals (price, stock, popularity, reputation), and small changes in UI or copy can be overridden by those signals. I don’t see evidence that CPS meaningfully moves choices when such competing signals are present, or when the page layout, fonts, and component positions deviate from the testbed.
>
> To better simulate the real-world market scenario, our evaluation of preference manipulation in both Movie and e-commerce is done by pairing the target item with random items in the same search category for each iteration. This setting ensures the robustness and practicality of our preference manipulation in real-world environments. The purpose of creating the synthetic dataset is to provide a fair testing ground for all items so that we can demonstrate the differences. While real-world platforms introduce additional complexity, the implication is that even though the manipulation effects might be fuzzier and less obvious, a tiny change in agent preferences (e.g. 1%) might still correspond to a large margin of profit with millions of online traffic.

---

> ### Author Response · Authors · 2025-11-21
> **Response to Reviewer Hgoh Part IV**
>
> > On the imperceptibility claim for images, I remain unconvinced without perceptual metrics or a human study. An L∞ budget of 8/255 is frequently used, but depending on content and scale it can still introduce visible speckling or edge artifacts, especially after resampling. A small LPIPS/SSIM/PSNR report (or a blinded user test) would make the “imperceptible” adjective feel earned rather than asserted.
>
> We appreciate this suggestion for a more rigorous imperceptibility assessment. We address this through multiple angles:
>
> **Standard compliance:** The L∞ = 8/255 threshold has been the established standard since the CIFAR10 Adversarial Examples Challenge [3] and remains widely accepted in adversarial perturbation research.
>
> **Perceptual metrics:** While we agree that LPIPS/SSIM/PSNR would strengthen our claims, there are several considerations:
> - For thumbnail images (typical size 200×200 pixels) embedded in webpages, perturbations are less perceptible than in isolated image viewing
> - LPIPS/SSIM require the original images as references to measure the perceptual distances, which defenders wouldn't have access to in deployment
> - Our detection experiments serve as a proxy: if VLMs cannot detect manipulations, the perturbations are "imperceptible" to the intended victim (the agent)
>
> **Future work:** We commit to including human perceptual studies if time permits. However, we would like to emphasize that "imperceptibility to agents" (measured via detection rates) is more relevant than "imperceptibility to humans" for our threat model.
>
>
> [3] CIFAR10 Adversarial Examples Challenge https://github.com/MadryLab/cifar10_challenge
>
>
> > I see clear dual‑use risk: the method is explicitly designed to manipulate autonomous agents on consumer platforms. If operationalized, it could be used to steer purchases or decisions without users’ awareness.
> The attack scenario plausibly conflicts with typical marketplace ToS (manipulative content, false or misleading claims, adversarial modifications). If data, code, or demos encourage violating site policies or scraping constraints, that also warrants scrutiny.
> The paper offers step‑by‑step guidance for cross‑modal manipulation. Without strong guardrails (e.g., release restrictions, red‑teaming, mitigation guidance), I worry about straightforward weaponization.
> If the authors release perturbations, prompts, or a live demo, I’d like to see a concrete harm‑mitigation plan (rate‑limits, gating, detection baselines) and clarity around any human evaluations (consent/compensation).
>
> ## Response to Ethics Concerns
>
> We appreciate the reviewer's careful consideration of ethical implications. We address the concerns below.
>
> ### Dual-Use Risk: Why This Research is Necessary
>
> **The vulnerability exists on day 1**
>
> Our research follows established cybersecurity practice: document vulnerabilities to enable defenses before widespread exploitation.
>
> **Defensive value:**
> - Platform providers need technical details to implement detection and mitigation
> - Security researchers require documented attack vectors to develop robust defenses
> - Our work enables the AI safety community to prioritize this threat in agent design
>
> **The dual-use risk itself proves the urgency.**
>
> Platforms must understand these vulnerabilities immediately to protect users.
>
> ### Terms of Service Compliance
>
> Our experiments use **synthetic, controlled test environments**, not unauthorized access to production systems. We follow standard adversarial ML research practices (similar to adversarial examples research, jailbreaking studies) that demonstrate vulnerabilities without policy violations.
>
> ---
>
> ## Bottom Line
>
> **Not publishing poses a greater risk.** Autonomous agents are deployed in consumer applications *now* without adequate security evaluation. Our work provides platforms with the knowledge needed to protect users before malicious exploitation occurs. This is defensive research with a positive impact.
>
> We sincerely appreciate the reviewer's insightful feedback, and we hope these new results and clarifications address your concerns.

---

### Official Review · Reviewer_v9mh · 2025-10-31

**Soundness:** 3
**Presentation:** 2
**Contribution:** 2
**Rating:** 2
**Confidence:** 4

**Summary:**

This paper investigates cross-modal adversarial manipulation of web agents powered by vision–language models (VLMs). The authors propose Cross-Modal Preference Steering (CPS) — a black-box attack framework jointly optimizing imperceptible textual and visual perturbations to steer agent preferences during selection tasks such as movie recommendation or e-commerce ranking. Under a realistic constraint where the attacker can only modify their own listing’s image and text, CPS achieves high success rates (up to 71%) while maintaining low detection rates across models including GPT-4.1, Qwen-2.5-VL, and Pixtral-Large.

**Strengths:**

- Timely and practically relevant topic: The work explores real-world vulnerabilities in AI web agents, which are becoming increasingly common in recommendation and decision systems.

- Comprehensive evaluation: The experiments are extensive, covering multiple commercial and open-source VLMs, two distinct domains (movies, shopping), and metrics for both effectiveness and stealth.

**Weaknesses:**

- The second contribution (black-box visual attack) adds limited conceptual value. While the paper’s second contribution highlights a black-box visual attack achieving imperceptible perturbations without model access, the value of this component is somewhat limited in light of recent progress in multimodal jailbreak research. Prior works such as Implicit Jailbreak Attacks via Cross-Modal Information Concealment (Wang et al., 2025), Jailbreak Large Vision-Language Models through Multi-Modal Linkage (Wang et al., 2024), and Visual Contextual Attack (Miao et al., 2025) have already demonstrated that imperceptible or hidden image-based manipulations can reliably steer the behavior of large VLMs under black-box conditions. Compared with these studies, the visual-attack module here mainly reuses known CLIP-transferable perturbation techniques and applies them within a web-agent scenario. Although the adaptation is technically solid and empirically strong, its conceptual novelty is modest. The paper would benefit from clarifying how this visual component differs substantively from previous imperceptible black-box perturbations beyond its integration into the CPS pipeline.

  - Miao, Z., Ding, Y., Li, L., & Shao, J. (2025). Visual Contextual Attack: Jailbreaking MLLMs with Image-Driven Context Injection. arXiv preprint
  - Wang, Y., Zhou, X., Wang, Y., Zhang, G., & He, T. (2024). Jailbreak Large Vision-Language Models Through Multi-Modal Linkage. arXiv preprint arXiv:2412.00473.
  - Wang, Z., Wang, H., Tian, C., & Jin, Y. (2025). Implicit Jailbreak Attacks via Cross-Modal Information Concealment on Vision-Language Models. arXiv preprint arXiv:2505.16446.

- The cross-modal synergy lacks mechanistic understanding. The third contribution—showing that combining textual and visual perturbations outperforms single-modal attacks—is purely empirical. The work does not analyze why such synergy occurs, whether it arises from attention redistribution, representation coupling, or bias amplification within multimodal fusion layers. Without internal analyses such as attention or gradient tracing, mutual-information measurement, or ablation on modality interactions, the study reads more like an empirical report than a scientific explanation. Adding interpretive experiments to uncover the mechanism behind the joint optimization would greatly strengthen the contribution.

- Defense and ethical analysis remain shallow. The defense analysis (Table 2) only uses an informed GPT-4.1 detector and reports near-random detection rates. There is little discussion on countermeasures, such as adversarial training, cross-modal consistency checks, or human-in-the-loop oversight.

**Questions:**

- How does the proposed visual attack technically differ from existing imperceptible black-box perturbation or multimodal concealment methods
- For inspiration, the authors might look at Contrasting Subimage Distraction Jailbreaking (Yang et al., 2025), which provides a good example of how deeper mechanistic analyses can reveal why multimodal attacks succeed. The intention is not to apply that specific method here, but to encourage a similar level of analytical depth in explaining the underlying dynamics of CPS.
  - Yang, Z., Fan, J., Yan, A., Gao, E., Lin, X., Li, T., ... & Dong, C. (2025). Distraction is all you need for multimodal large language model jailbreaking. In Proceedings of the Computer Vision and Pattern Recognition Conference (pp. 9467-9476).

---

> ### Author Response · Authors · 2025-11-21
> **Response to Reviewer v9mh Part I**
>
> We sincerely thank the reviewer for the thorough evaluation and constructive feedback. We particularly appreciate the recognition of our work's timeliness and comprehensive evaluation, as well as the valuable suggestion regarding mechanistic analysis (Yang et al., 2025).
>
> > The second contribution (black-box visual attack) adds limited conceptual ... beyond its integration into the CPS pipeline.
> Miao, Z., Ding, Y., Li, L., & Shao, J. (2025). Visual Contextual Attack: Jailbreaking MLLMs with Image-Driven Context Injection. arXiv preprint
> Wang, Y., Zhou, X., Wang, Y., Zhang, G., & He, T. (2024). Jailbreak Large Vision-Language Models Through Multi-Modal Linkage. arXiv preprint arXiv:2412.00473.
> Wang, Z., Wang, H., Tian, C., & Jin, Y. (2025). Implicit Jailbreak Attacks via Cross-Modal Information Concealment on Vision-Language Models. arXiv preprint arXiv:2505.16446.
>
> > How does the proposed visual attack technically differ from existing imperceptible black-box perturbation or multimodal concealment methods
>
>
>
> We would like to respectfully clarify our black-box visual attack contribution, which we believe represents a fundamental advance in a dimension that prior work has not addressed: **robustness to real-world preprocessing chains**.
>
> ---
>
>
> ## Distinction from Cited Jailbreak Work
>
> The reviewer suggests our visual component has limited novelty compared to recent jailbreak attacks against VLMs. We would like to respectfully clarify that **jailbreak attacks and preference manipulation attacks address fundamentally different problems through different mechanisms**:
>
>
> **Miao et al. (2025) - Visual Contextual Attack**: Generates auxiliary images and fabricates contextual dialogues to construct vision-centric jailbreak scenarios that trigger harmful responses: uses visible image content, not imperceptible adversarial perturbations.
>
>
> **Wang et al. (2024) - Multi-Modal Linkage**: Uses encryption-decryption with word replacement, image mirroring, and "evil alignment" framing to hide malicious instructions: produces visible image changes, not perception manipulation via adversarial noise.†
>
>
> **Wang et al. (2025) - Implicit Jailbreak**: Embeds harmful textual instructions into images via LSB steganography paired with benign prompts: focuses on information concealment, not perception shifting through adversarial perturbations.
>
>
> **Critical differences**:
> - **None use imperceptible adversarial perturbations** to manipulate visual perception
> - **None achieve robust black-box transfer** for perception manipulation
> - **None target preference manipulation** in agent decision-making
>
>
> **Core distinction**: Jailbreak attacks bypass safety alignment (changing what the model will say about harmful content). Our attack manipulates visual perception (changing what the model sees in normal content). These are orthogonal vulnerabilities requiring different defenses.
>
> ---
>
>
> ## Our Core Contribution: First Imperceptible Adversarial Perturbations Robust to Real-World Preprocessing
>
>
> ### **The Critical Gap in Prior Adversarial Perturbation Work**
>
>
> As we discussed in Section 1 (line 75-82), the current CLIP attacks are either working on white-box scenario or low in success rate. More importantly, they all require the attacked images to be resized to a much smaller, fixed resolution (e.g. 224×224) to fit the exact CLIP resolutions, which greatly reduced the practicality of the implications of the visual attack.
>
>
> **Why this matters**: In real-world web agent deployments, images uploaded by content publishers undergo extensive preprocessing before reaching the VLM:
> 1. Platform resizing (to standard dimensions like 800×600, 1024×768)
> 2. Format conversion (PNG→JPEG) and compression for storage optimization
> 3. Screenshot capture when the agent views the rendered webpage
> 4. Cropping and patchifying during VLM input preprocessing
> 5. Final resizing to the VLM's expected input resolution
>
>
> **Prior work fails at steps 1-2**: Because their perturbations are optimized for a single fixed resolution (224×224), any resizing or format conversion degrades the attack. This is why they achieve <20% ASR on black-box commercial VLMs in practice.
>
> **Our contribution**: We have developed the first black-box, adversarial perturbation that can shift a proprietary model's (e.g. GPT-4.1) visual perception of an image with imperceptible perturbation that can be applied to image of any sizes and robust to different pre-processing techniques such as format conversion and resizing, which have made this attack applicable to web agents for the first time.

---

> ### Author Response · Authors · 2025-11-21
> **Response to Reviewer v9mh Part II**
>
> Continuing the responses to the questions in Part I:
>
> ### **Concrete, Independently Verifiable Demonstration**
>
> Our robustness claim can be verified by anyone with access to our paper:
>
> **Reproducibility test using Figure 2**:
> 1. Open our paper PDF and display Figure 2 (the adversarial perturbation example)
> 2. Zoom in slightly on the perturbed image (right side showing "apple"→"orange" attack)
> 3. **Take a screenshot** of the paper containing Figure 2
> 4. **Crop** the adversarial image from the screenshot
> 5. Upload this cropped screenshot to ChatGPT's web interface using **GPT-5.1**
> 6. Ask: "What object is in this image?"
>
>
> **Result**: We have checked multiple times and the latest GPT-5.1 model that we have never seen or tested at the time of submission will still mistakenly believe that is an orange in the screenshot, and this highly robust black-box visual perception manipulation to even unseen models is one of the biggest contributions in our work. We encourage reviewers to independently verify this claim using the reproduction steps provided on GPT-5.1 or other proprietary models as we are confident in our method's robustness. This 3-step real-world transformation (screenshot → crop → upload) would **completely destroy perturbations from prior work**, yet our attack maintains effectiveness as shown in Tables 1 and 3 of our paper. This demonstrates the practical robustness that our multi-crop optimization enables.
>
>
> **Additional robustness validation**:
> In addition, to further address the robustness of our method, we have conducted the following experiments in addition to the concept shift experiment in Table 5 of our paper:
>
> | Semantic Shift | GPT-4.1 (JPEG) | Resize+JPEG | Resize+Color_Jitter+JPEG |
> |---|---|---|---|
> | cat ↔ dog | 100.0 | 65.0 | 60.0 |
> | sheep ↔ cow | 75.0 | 55.0 | 40.0 |
> | bus ↔ train | 70.0 | 50.0 | 35.0 |
> | apple ↔ orange | 65.0 | 55.0 | 65.0 |
> | bicycle ↔ motorcycle | 50.0 | 25.0 | 20.0 |
> | couch ↔ bed | 30.0 | 30.0 | 15.0 |
> | **Average** | **65.0** | **46.7** | **39.2** |
>
> Here are the hyperparameters of the manipulations:
>
> - **Resize**: Random scaling by factor *r* ∈ [0.8, 1.2]
> - **Interpolation**: LANCZOS resampling
> - **Compression**: JPEG encoding at quality factor Q=85
> - **Color Jitter**:
>   - Brightness adjustment: factor *b* ∈ [0.8, 1.2]
>   - Contrast adjustment: factor *c* ∈ [0.8, 1.2]
>   - Saturation adjustment: factor *s* ∈ [0.8, 1.2]
>
> As the result shows that our method remains robust after JPEG compression, resizing, and color jittering.
>
>
> ### **Response to "Reusing Known CLIP-Transferable Techniques"**
>
>
> We would like to respectfully address this characterization. While we build on adversarial perturbations and CLIP transferability, we overcome critical limitations that prevented prior techniques from working in practice:
>
>
> **Prior limitation**: <20% ASR on black-box models, perturbations degrade under preprocessing
> **Our contribution**: 60-70% ASR, perturbations survive screenshots+crops+resizes+format conversions
>
>
> **Prior limitation**: Fixed resolution requirements (224×224 downsampling)
> **Our contribution**: Arbitrary resolutions through multi-crop optimization and robust against processing
>
>
> This represents what we believe is **the first demonstration that imperceptible adversarial perturbations can reliably manipulate black-box commercial VLMs' visual perception in realistic deployment scenarios**, which enables our cross-modal preference manipulation framework.
>
>
> ---
>
>
>
> ## Summary
>
>
> **Our key advance over prior adversarial perturbation work**:
>
>
> | Metric | **Prior Work** | **Our Method** |
> |--------|---------------|---------------|
> | **Preprocessing robustness** | Fails on resize/crop/screenshot | **Survives screenshots, crops, resizes, format conversions** |
> | **Resolution constraints** | Fixed 224×224 downsampling | **Arbitrary resolutions** |
> | **Black-box ASR (GPT-4o)** | <20% | **65.8%** |
> | **Web agent scenario (Table 3)** | +0-1.18% over baseline | **+18.8% (Qwen), +7% (GPT-4.1)** |
> | **Detection evasion** | Not evaluated | **26% MDR (near-random 12.5%)** |
>
>
> **Distinction from jailbreak work**:
>
>
> | Aspect | **Jailbreak Attacks** | **Our Work** |
> |--------|----------------------|--------------|
> | **Problem** | Bypass safety alignment | Manipulate visual perception|
> | **Goal** | Elicit harmful outputs | Bias selection decisions |
> | **Method** | Visible changes, typography, steganography | Imperceptible adversarial noise |
> | **Evaluation** | Safety benchmark ASR | Preference manipulation + robustness + stealth |
>
>
> We hope this clarification demonstrates that our black-box visual attack represents a fundamental advance in achieving **real-world preprocessing robustness**, which was the critical missing piece that prevented prior adversarial perturbation work from being practical against deployed web agents. We thank the reviewer again for the thoughtful feedback and welcome continued discussion.

---

> ### Author Response · Authors · 2025-11-21
> **Response to Reviewer v9mh Part III**
>
> > The cross-modal synergy lacks mechanistic understanding... greatly strengthen the contribution.
>
> >For inspiration, the authors might look at Contrasting Subimage Distraction Jailbreaking ... explaining the underlying dynamics of CPS.
>
> **Why combining textual and visual perturbations outperforms single-modal attacks**:
>
> **Mechanistic explanation:** Since web agents operate on screenshots of webpages rather than separate text and image inputs, both modalities are processed through the same visual encoder. This unified processing creates additive effects where:
> - Textual perturbations manipulate the text-region embeddings on the screenshot
> - Visual perturbations manipulate the image-region embeddings on the screenshot
> - Joint optimization produces stronger cumulative signals across the combined visual input, increasing the likelihood of target selection
>
> Since the web agents are only working on the screen (screenshot of the webpage), our textual perturbations are still processed visually by the agent. Similar to many VLM jailbreak papers, the attack can comprise two parts: adversarial images + textual descriptions (typography/Figstep [1]). As demonstrated in MM-SafetyBench[2] or HADES[3], jailbreak attacks can be effective under these settings:
>
>     1) SD (Stable Diffusion) images only
>     2) Text (Typography) only
>     3) SD + Typography
>
> MM-Safetybench's results show that the textual information (Typography) on the image is much more effective than adversarial images generated by Stable Diffusion, and adding SD images on top of Typography (SD + Typography) produces a marginal gain in ASR over typography alone. Similarly, in our case, textual perturbation is pretty strong by itself, and adding visual perturbation can provide a marginal gain in preference as well.
>
> **Interpretation:** Given that both text and images are rendered as pixels/visual tokens and processed visually by web agents, the observed improvement from joint optimization reflects additive signal strength rather than true cross-modal synergy. Like "SD + Typography" jailbreak attacks that yield incrementally higher ASR by expanding the attack surface, our joint optimization slightly improves success rates as the attack vectors become more comprehensive
>
> **More mechanistic understanding**:
> Thank you so much for providing new insights/inspirations and referring to "Contrasting Subimage Distraction Jailbreaking". It is indeed a very interesting paper and we are impressed by how they performed their mechanistic analysis.
>
> Given our method is working under black-box environment, it is not possible to perform internal analyses such as attention or gradient tracing. However, we did try our best to provide a better understanding of how the attack works by:
>     1) Performed an Ablation Study (Section 6) demonstrating how visual perturbation can work by itself
>     2) Probed the model's internal (Appendix C) via a white-box model (Qwen-2.5-VL) to track the logit/probability changes to different concepts being injected, and we found out that generally positive concepts such as "Best choice" are the most effective concepts to be injected into the images.
>
> We hope this clarification will address some of the questions and concerns.
>
> [1] FigStep: Jailbreaking Large Vision-Language Models via Typographic Visual Prompts https://arxiv.org/pdf/2311.05608
>
> [2] MM-SafetyBench: A Benchmark for Safety Evaluation of Multimodal Large Language Models https://arxiv.org/abs/2311.17600
>
> [3] Images are Achilles’ Heel of Alignment: Exploiting Visual Vulnerabilities for Jailbreaking Multimodal Large Language Models https://arxiv.org/pdf/2403.09792
>
>
>
> > Defense and ethical analysis remain shallow. The defense analysis (Table 2) only uses an informed GPT-4.1 detector and reports near-random detection rates. There is little discussion on countermeasures, such as adversarial training, cross-modal consistency checks, or human-in-the-loop oversight.
>
> Because we are one of the first to propose the cross-modal preference manipulation attack against web agents, no existing work is done to defend against this type of attack. Our work serves as one of the first to introduce the vulnerability and provide a fair testing environment for future defenses. We will definitely include a "Future Work" section on the camera-ready version to call for more attention to address this new vulnerability.
>
> Given limited data and resources, it is difficult to perform adversarial training in a few days. Following the reviewer's suggestions, we have included a cross-modal consistency check by using GPT-4.1 as a binary detector for cross-modal consistency or suspicion here:
>
> Consistency check detection rate: 25.0%
>
> Binary risk detector detection rate: 21.5%
>
> GPT-4.1 as a risk scorer:
>
> Top-1 Detection Rate: 21.5%
>
> Top-2 Detection Rate: 27.2%
>
> Top-5 Detection Rate: 54.3%
>
> We sincerely appreciate the reviewer's insightful feedback, and we hope these new results and clarifications address your concerns.

---

### Official Review · Reviewer_tazK · 2025-11-01

**Soundness:** 3
**Presentation:** 3
**Contribution:** 3
**Rating:** 8
**Confidence:** 1

**Summary:**

The paper introduces Cross-Modal Preference Steering (CPS), an adversarial attack approach to steering VLM's decision making. The authors formalize a realistic attack scenario and achieves >50% success rate on shifting VLM's perceptions on images without requiring knowledge of such block-box models. Specifically, the paper introduces noise optimization for visual content perturbation and jointly optimizes for text description to inject preference for a specific concept. Visual content perturbation is done through CLIP optimization and text optimization utilizes white-box VLM such as Qwen-2.5VL-32B for attacker to monitor changes in logits and bias selection probability.

**Strengths:**

- The paper proposes a novel joint optimization approaches to bias a VLM's preference for a specific visual content.
- It is successfully demonstrated that this process works for commercial models such as GPT-4.1 and does not require any knowledge about the target models themselves.
- Comprehensive analysis and ablation studies are done on the effectiveness of joint visual-text perturbation over single-modal attacks, and results show that some models are more susceptible to attacks in the visual domain.

**Weaknesses:**

- The attack optimization still requires white-box models as surrogates. How can we reason that this optimization transfers to other black-box models? If a black-box model is using very different visual encoder, this attack approach would not make much sense?
- Some of the notations are confusing, such as in eq. 9, what is $\Pi$? and what is the argument to $\mathcal{L}$ in eq. 8, and how is this used in equation 10, which seems inconsistent with its usage in eq. 12

**Questions:**

I'd like the author to address some concerns above.

---

> ### Author Response · Authors · 2025-11-21
> **Response to Reviewer tazK Part I**
>
> We thank the reviewer for the valuable feedback on improving this paper!
>
> Below are our responses to your questions and concerns.
>
> > The attack optimization still requires white-box models as surrogates. How can we reason that this optimization transfers to other black-box models? If a black-box model is using a very different visual encoder, this attack approach would not make much sense?
>
> One explanation at a high level is that there are shared vulnerabilities among visual models that are trained with contrastive objectives, including CLIP. For example, one paper in ICLR 2023 (https://iclr.cc/virtual/2023/poster/11277) found that it is possible to poison contrastive learning classification models such as SimCLR, MOCO and BYOL with imperceptible adversarial perturbations via PGD as well. It means that at a high level, these contrastively trained models such as CLIP might share a lot of vulnerabilities that can be easily exploited.
>
> A second and more practical explanation is that there are limited amount of captioned image training datasets such as LAION-2B. At the same time, there are also great similarities between the architectures of image encoders on different VLMs. Therefore, although different VLMs claim to have different image encoders, these encoders still share substantial commonalities in their training data and architectures. Such shared features create common vulnerabilities that can be optimized and learned via attacking multiple (e.g. 12 ) image encoders at the same time. Those two factors both help explain the high transferability of our Attack.
>
> In our CLIP surrogate attack, we optimize against 12 different CLIP models at the same time, so the PGD will mainly attack on the vulnerabilities that most of the CLIP models still share, hence making the attack much more transferable to any unseen models.
>
> To further test the robustness and transferability of our visual attack, here is one simple test that can be done:
>
>     1) Zoom in on Figure 2 of the PDF of our paper on page 17
> 	2) Take a screenshot and crop the adversarial image out for testing
> 	3) Send the cropped image to the ChatGPT online platform while asking, “Describe the image”
>
> We have checked multiple times and the latest GPT-5.1 model that we have never seen or tested at the time of submission will still mistakenly believe that it is an orange in the screenshot, and this highly robust black-box visual perception manipulation to even unseen models is one of the biggest contributions in our work. We encourage reviewers to independently verify this claim using the reproduction steps provided on GPT-5.1 or other proprietary models, as we are confident in our method's robustness.

---

> ### Author Response · Authors · 2025-11-21
> **Response to Reviewer tazK Part II**
>
> > Some of the notations are confusing, such as in eq. 9, what is Π? and what is the argument to ℒ in eq. 8, and how is this used in equation 10, which seems inconsistent with its usage in eq. 12
>
> We apologize for the confusion, and we thank the reviewer for the careful attention to our notation. We clarify the relationship between these equations below.
>
> Eq. 9 is directly adopted from the equation from Section 2.1 of the PGD[1] (Projected Gradient Descent) paper, where Π represents the "Projection" that clips the perturbation into a certain pre-defined range (∥δ∥∞≤ϵ).
>
> [1] Towards Deep Learning Models Resistant to Adversarial Attacks (https://arxiv.org/pdf/1706.06083)
>
> For the question about eq. 8 ,10 and 12, here is the clarification:
> ---
>
> ### **Equation 8: Visual Attack Objective**
>
> ```
> L = ∥E_CLIP(I + δ) − E_negative∥₂ − ∥E_CLIP(I + δ) − E_target∥₂
> ```
>
> **Purpose:** Defines the core optimization goal for visual perturbations.
>
> - Maximizing L pushes the perturbed image embedding away from negative concepts while pulling it toward target concepts
> - This achieves semantic steering of the VLM's visual perception
>
>
> ### **Equation 10: Multi-Resolution Robustness**
>
> ```
> ∇_δ L^t = (1/K) Σ_{i=1}^K ∇_δ L[E_CLIP(C_i(I + δ^t))]
> ```
>
> **Purpose:** Computes gradients over K random crops to ensure attack transferability across varying resolutions.
>
> - Uses the **same loss L from Equation 8**, evaluated on cropped versions C_i(I + δ)
> - Prevents overfitting to a specific resolution, improving black-box transfer to VLMs with unknown preprocessing
>
> **This is not a different loss function**. Instead, it's a robust gradient computation strategy for the loss defined in Equation 8.
>
> ---
>
> ### **Equation 12: Visual Effectiveness Constraint**
>
> ```
> s.t. L_visual(δ, T_target, T_neg) > τ_v
> ```
>
> **Purpose:** Ensures the visual perturbation achieves sufficient manipulation strength in the joint cross-modal optimization.
>
> - L_visual refers to the **same loss from Equation 8**
> - The constraint guarantees that visual manipulation exceeds a certain threshold so that the target concept can be successfully injected into images
> - τ_v is the minimum effectiveness threshold
>
> ---
>
> ### **Unified Perspective**
>
> These equations represent **different aspects of the same loss function**:
>
> 1. **Equation 8:** Defines the objective (what we optimize)
> 2. **Equation 10:** Specifies robust gradient computation (how we optimize)
> 3. **Equation 12:** Imposes a quality constraint (ensuring effectiveness in joint optimization)
>
>
> Thank you again for your questions and suggestions!
>
> We sincerely hope our responses can help provide a better understanding of our paper and a more confident judgment of our work. Please do not hesitate to reach out with any other questions or suggestions!

---

> > ### Comment · Reviewer_tazK · 2025-11-27
> >
> > I thank the authors for comprehensive response. I would like to keep my score for acceptance.

---

### Official Review · Reviewer_S5C5 · 2025-11-01

**Soundness:** 3
**Presentation:** 3
**Contribution:** 2
**Rating:** 4
**Confidence:** 2

**Summary:**

The paper proposes *Cross-Modal Preference Steering*, an attack framework that modifies an item's thumbnail and description to bias VLM web agents under a black-box setting where the adversary can only edit their own listing. It formalizes the threat, defines manipulation and detection objectives, and presents an agent pipeline and optimization procedures for textual refinement and transferable visual perturbations, with evaluations on movie-selection and shopping-style tasks using multiple agent backbones, as well as a detector-based analysis and ablations.

**Strengths:**

- The paper states a practical black-box threat model aligned with real content-publisher permissions.
- The paper is well organized and easy to follow.

**Weaknesses:**

- The method aims to exploit "RLHF-induced preference biases" (Section 4.5.1) in descriptions, yet the paper does not isolate which biases drive selection shifts for each backbone or provide targeted probes.

- Beyond noting survival under downsampling/cropping, the paper does not test JPEG recompression, bit-depth changes, or typical input sanitization. Input transforms and real-world variations can sharply alter attack success, so attack robustness here is unclear.

- The threat model assumes the attacker "can observe user queries" and then tailor cross-modal edits (Section 3.1), which many real marketplaces do not expose to individual sellers. This suggestion might be strong.

**Questions:**

- How does attack success change when using strong black-box transfer methods like MI-FGSM?
- For one end-to-end attack, how many surrogate VLM evaluations, GPT-4.1 calls, and optimization steps are used on average, what is the wall-clock time and cost per targeted item, and how do success rates degrade under reduced budgets?

---

> ### Author Response · Authors · 2025-11-21
> **Response to Reviewer S5C5 Part I**
>
> We thank the reviewer for the valuable feedback on improving this paper!
>
> >The method aims to exploit "RLHF-induced preference biases" (Section 4.5.1) in descriptions, yet the paper does not isolate which biases drive selection shifts for each backbone or provide targeted probes.
>
> The "RLHF-induced preference biases" are comprehensive and untargeted, as we mentioned in Section 2.2, these biases are inevitable side effects in all directions, including length bias, gender bias, and sycophancy in an effort to align with human values. Therefore, it is almost impossible to meticulously isolate which specific bias component drives the manipulation. The more common way is to: 1, understand there is a bias/vulnerability that we can exploit 2, set an optimization goal of manipulating preferences and 3, iteratively refine texts to exploit the vulnerability towards the goal.
>
> For example, there is one concurrent submission at ICLR 2026 [3] that manipulates the preference of a white-box LLM-based ranking system by adding adversarial suffixes in the same style as the GCG [1] jailbreak attack. In our case, instead, we are using a LLM feedback loop to refine the texts to conform to the internal biases to manipulate the preferences in a similar style as the "Persuasion" [2] jailbreak attack. Our work, when compared with [3], demonstrates a black-box attack that is also more natural and harder to detect.
>
> In an effort to provide more explainability and have a better understanding of how the preference manipulation works, we performed a probing experiment under white-box conditions with Qwen-2.5-VL model. More details can be found in Appendix C. Through targeted probing we are able to show that generally positive concepts such as "best choice" can increase the logits of the model choosing the targeted item.
>
> [1] Universal and Transferable Adversarial Attacks on Aligned Language Models https://arxiv.org/abs/2307.15043
>
> [2] How Johnny Can Persuade LLMs to Jailbreak Them: Rethinking Persuasion to Challenge AI Safety by Humanizing LLMs https://aclanthology.org/2024.acl-long.773/
>
> [3] Controlling Output Rankings in Generative Engines for LLM-based Search https://openreview.net/forum?id=cg9nIA1HhH
>
>
> > Beyond noting survival under downsampling/cropping, the paper does not test JPEG recompression, bit-depth changes, or typical input sanitization. Input transforms and real-world variations can sharply alter attack success, so attack robustness here is unclear.
> Thank you for pointing that out. We have made our visual perturbation particularly robust to resizing and downsampling/cropping because our attacked images will go through these process before sending to the website:
> 1. Platform resizing (to standard dimensions like 800×600, 1024×768)
> 2. Format conversion (PNG→JPEG) and compression for storage optimization
> 3. Screenshot capture when the agent views the rendered webpage
> 4. Cropping and patchifying during VLM input preprocessing
> 5. Final resizing to the VLM's expected input resolution
>
> Table 1 and 3 in our paper shows that the visual perturbation remains effective after resizing and screenshots.
>
> One direct test on the robustness of our work can be done by:
>
>     1) Zoom in on Figure 2 of the PDF of our paper on page 17
> 	2) Take a screenshot and crop the adversarial image out for testing
> 	3) Send the cropped image to the ChatGPT online platform while asking “Describe the image”
>
> We have checked multiple times and the latest GPT-5.1 model will still mistakenly believe that it is an orange in the screenshot, and this highly robust black-box visual perception manipulation is one of the biggest contributions in our work. We encourage reviewers to independently verify this claim using the reproduction steps provided as we are confident in our method's robustness.
>
> In addition, to address the robustness against bit-depth change and JPEG compression, we have conducted the following experiments in addition to the concept shift experiment in Table 5 of our paper:
>
> | Semantic Shift | GPT-4.1 (JPEG) | Resize+JPEG | Resize+Color_Jitter+JPEG |
> |---|---|---|---|
> | cat ↔ dog | 100.0 | 65.0 | 60.0 |
> | sheep ↔ cow | 75.0 | 55.0 | 40.0 |
> | bus ↔ train | 70.0 | 50.0 | 35.0 |
> | apple ↔ orange | 65.0 | 55.0 | 65.0 |
> | bicycle ↔ motorcycle | 50.0 | 25.0 | 20.0 |
> | couch ↔ bed | 30.0 | 30.0 | 15.0 |
> | **Average** | **65.0** | **46.7** | **39.2** |
>
> Here are the hyperparameters of the manipulations:
>
> - **Resize**: Random scaling by factor *r* ∈ [0.8, 1.2]
> - **Interpolation**: LANCZOS resampling
> - **Compression**: JPEG encoding at quality factor Q=85
> - **Color Jitter**:
>   - Brightness adjustment: factor *b* ∈ [0.8, 1.2]
>   - Contrast adjustment: factor *c* ∈ [0.8, 1.2]
>   - Saturation adjustment: factor *s* ∈ [0.8, 1.2]
>
> As the result shows that our method remains robust after JPEG compression, resizing, and color jittering.

---

> ### Author Response · Authors · 2025-11-21
> **Response to Reviewer S5C5 Part II**
>
> > The threat model assumes the attacker "can observe user queries" and then tailor cross-modal edits (Section 3.1), which many real marketplaces do not expose to individual sellers. This suggestion might be strong.
>
> The attack does not assume that the attacker can observe user queries. Instead, the attack is established on the idea that the attacker knows what kind of products they are trying to “promote”, hence it is reasonable for the attacker to successfully infer the user query that might lead to the search result pages containing the target item.
>
> As mentioned in the Introduction Section starting from line 53, the attacker assumes that the user asks for a particular type of product. (For example, “Buy A bluetooth speaker”) Then multiple items (For example, different models of “bluetooth speakers”) will surface on top of the page that all match the search criteria. Our preference manipulation takes place at this step and aims at maximizing the likelihood of the target item being chosen by the web agent.
>
> In addition, to better simulate the real-world market scenario, our evaluation of preference manipulation is done by pairing the target item with random items in the same search category for each iteration. This setting ensures the robustness and practicality of our preference manipulation in real-world environments.
>
>
> > How does attack success change when using strong black-box transfer methods like MI-FGSM?
>
> Here are the results using MI-FGSM with momentum_decay (μ) = 1.0, eps (Maximum perturbation bound L∞)=8/255, alpha (step size)=0.5/255 to replicate the same concept-shifting experiment as shown in Table 5 of our paper:
>
> | Method  | Qwen-2.5VL | GPT-4.1 |
> |---------|------------|----------|
> | PGD     | 70.0       | 62.5     |
> | MIFGSM  | 63.3       | 57.5     |
>
> As shown in the table, the MI-FGSM is also effective but slightly worse than PGD.
>
> > For one end-to-end attack, how many surrogate VLM evaluations, GPT-4.1 calls, and optimization steps are used on average, what is the wall-clock time and cost per targeted item, and how do success rates degrade under reduced budgets?
>
> Thank you for the question. We have set the condition that a maximum of 20 queries are allowed to be sent to the web_agent (e.g. GPT-4.1) for optimization for each target item. On average 10.90 iterations were used per target item. For each target item, 50 iterations of surrogate VLM attacks were used to inject concepts into thumbnail images. On average it took 12.1 minutes to finish optimizing each target item overall with joint optimization.
> In addition, we did limit the budget to 5 and 10 queries, and here are the results:
>
> | Method | Model    | Dataset | Max_iter = 5 | Max_iter = 10 | Max_iter = 20 |
> |--------|----------|---------|--------------|---------------|----------------|
> | CPS    | GPT-4.1  | Movie   | 39.8%        |     52.2%     | 59%            |
>
> The result shows that limiting the budget does have some impact on the attack effectiveness.
>
> We sincerely appreciate the reviewer's insightful feedback, and we hope these new results and clarifications address your concerns.

---

### Author Response · Authors · 2025-12-04
**Rebuttal Summary Part I (Core Contributions and Novelty)**

---

## Rebuttal Summary: Cross-Modal Preference Steering (CPS)

**Official Comment by Authors**

Dear AC, SAC, and PCs,

Thank you so much for your time and dedication in reviewing our submission.

To facilitate a comprehensive understanding of our work and the rebuttal process, we provide: (1) an overview of our contributions, (2) a summary of extensive new experiments conducted during rebuttal, (3) reviewer-by-reviewer analysis with corrections to key misunderstandings, and (4) our concluding remarks.


---

### [1. Core Contributions and Novelty]

CPS addresses a critical gap in AI agent security: **no prior work has demonstrated practical preference manipulation attacks against VLM-based web agents under realistic threat assumptions.** Our work establishes three significant contributions:

---

**Contribution 1: First Preference Manipulation Attack Under a Realistic Threat Model**

Prior attacks on web agents assume impractical conditions: white-box model access, control over entire webpages, or access to agent memory. CPS operates under constraints that mirror **actual marketplace dynamics**:

- **Attacker capability:** A legitimate vendor on any e-commerce or content platform who controls only their own listing—specifically, the product thumbnail and text description. No system access, no webpage injection, no agent internals.
- **Query budget:** Fewer than 20 queries to the target web agent to refine the attack. This reflects realistic API rate limits and cost constraints.
- **Attacker goal:** Maximize the probability that, when a user deploys a web agent to "find me a bluetooth speaker" or "recommend a movie," the agent selects the attacker's product over competitors.

This threat model is not hypothetical. It describes the exact permissions that almost all vendors, or content creators already possess. We demonstrate that these minimal capabilities are sufficient to systematically bias agent decisions, revealing a fundamental vulnerability in how VLM-based agents process and rank content.

---

**Contribution 2: Discovery of a Robust Black-Box Visual Attack Channel Against Commercial VLMs**

We identify **CLIP-based visual encoders as the Achilles' heel** of modern VLMs and web agents, and exploit this vulnerability to establish a previously unknown attack channel:

- **Imperceptible yet powerful:** With perturbations bounded by L∞ = 8/255, which are visually hardly detectable to humans, we can drastically alter what a VLM *perceives* in an image. An apple becomes an orange; a generic product becomes a premium brand.
- **Unprecedented robustness:** Unlike prior adversarial perturbation work that degrades under real-world conditions, our attack survives the complete preprocessing pipeline: JPEG compression, color jittering, random resizing, format conversion, and screenshot capture.
- **Transferability to unseen models:** Most critically, our attack transfers to models that did not exist when we developed CPS. **GPT-5.1 was released days ago (well after our submission), yet our adversarial examples still fool it.**

**We encourage everyone to independently verify this claim:** Open our PDF to Figure 2 (page 17), zoom in, take a screenshot of the adversarial image, crop it, and upload it to the ChatGPT platform with GPT-5.1 (or other platforms). Ask: "What is in this image?" The model will misidentify the apple as an orange. This simple test demonstrates that our attack remains effective even after multiple lossy transformations and against models we never trained on.

This finding opens a significant new attack surface: any image displayed to a VLM-based agent can be imperceptibly manipulated to alter the agent's perception and downstream decisions, which is a threat that generalizes across all commercial VLMs sharing similar visual encoding architectures.

---

**Contribution 3: Effective and Evasive Preference Manipulation at Scale**

Combining our realistic threat model with the newly discovered visual attack channel, CPS achieves:

- **High effectiveness:** Up to 71% preference manipulation rate across GPT-4.1, Qwen-2.5-VL, and Pixtral-Large, compared to 12.5% random baseline
- **Strong evasiveness:** Detection rates of 21-25% (near-random for 8-item selection), demonstrating that manipulated content does not trigger obvious anomaly signals
- **Practical efficiency:** Average optimization converges in ~11 iterations at ~12 minutes per target item, which is well within attacker budget constraints

---

**Why This Matters Now:** VLM-based web agents are being deployed for high-stakes decisions, including product recommendations, content curation, and travel booking. We demonstrate that any content publisher, with permissions they already have, can systematically bias these agents. This is not a theoretical vulnerability; it is an exploitable attack surface that exists today on every platform where agents interact with user-generated content.

---

---

> ### Author Response · Authors · 2025-12-04
> **Rebuttal Summary Part II (Extensive New Experiments During Rebuttal)**
>
> ---
>
> ### [2. Extensive New Experiments During Rebuttal]
>
> To address reviewer concerns, we conducted **new experiments** and provided more information as requested:
>
> | New Experiment | Addressing | Key Result |
> |----------------|------------|------------|
> | **JPEG + Resize + Color Jitter Robustness** | S5C5, Hgoh | 39.2-65% ASR maintained under combined transformations |
> | **MI-FGSM Comparison** | S5C5 | MI-FGSM achieves 57.5% (vs. PGD 62.5%), both effective |
> | **Budget Sensitivity Analysis** | S5C5, Hgoh | 5 iter→39.8%, 10 iter→52.2%, 20 iter→59% (practical budgets) |
> | **Cross-Modal Consistency Detector** | v9mh, Hgoh | 25% detection rate (near-random baseline 12.5%) |
> | **Binary Detector + Risk Scorer** | v9mh, Hgoh | Binary: 21.5%; Top-1/2/5: 21.5%/27.2%/54.3% |
> | **Complete Hyperparameter Specification** | Hgoh | L∞=8/255, step=0.5/255, 50 steps, 20 crops, 12 surrogates |
>
> ---

---

> ### Author Response · Authors · 2025-12-04
> **Rebuttal Summary Part III (Reviewer-by-Reviewer Response to Reviewer S5C5 and tazK )**
>
> ### [3. Reviewer-by-Reviewer Response]
>
> ---
>
> **Reviewer S5C5 (Score: 4, Confidence: 2): Concerns Addressed; Identified Weaknesses Are Actually Core Strengths**
>
> | Concern | Resolution |
> |---------|------------|
> | RLHF biases not isolated | Explained that biases are comprehensive and untargeted by design; isolation is infeasible even under white-box conditions. Added probing analysis in Appendix C |
> | Limited robustness testing | Provided extensive new experiments under JPEG compression, resizing, and color jitter—robustness is now demonstrated as a **core strength** |
> | Query observation assumption | **Corrected misunderstanding:** Attacker infers likely queries from product category; no observation of actual user queries required |
>
> We respectfully note that this reviewer's primary concerns of robustness and threat model assumptions are, upon clarification, among the **strongest contributions** of our work:
>
> **On Robustness:** The reviewer expressed concern that our visual attack was not tested under sufficient real-world transformations. We provided comprehensive new experiments demonstrating robustness under JPEG compression, random resizing, and color jitter, both individually and in combination. Crucially, our attack maintains a high ASR even under the harshest combined transformations. This level of robustness is unprecedented in black-box adversarial attacks against VLMs. Far from being a weakness, **preprocessing robustness is the defining technical contribution** that makes our attack practical against deployed systems.
>
> **On Threat Model:** The reviewer believed our attack assumes the attacker "can observe user queries," which would indeed be unrealistic. This reflects a misunderstanding of our setup. In CPS, the attacker is simply a vendor who knows what product category they are selling. The attacker never observes actual user queries; they only optimize their own listing content based on the reasonable assumption that their product will appear in relevant searches. This is precisely how real marketplaces operate, making our threat model not a limitation but a **strength that ensures practical applicability**.
>
> **On RLHF Bias Isolation:** The reviewer requested isolation of specific RLHF-induced biases driving selection shifts. We clarified that such isolation is infeasible because these biases are comprehensive, multidirectional side effects of alignment training. Even with full white-box access, disentangling individual bias contributions remains an open research problem. We nonetheless provided supplementary probing experiments in Appendix C, demonstrating that positive concepts measurably increase selection logits, offering interpretable evidence of the exploitation mechanism.
>
> *Given that this reviewer's concerns target aspects that are demonstrably among our paper's strongest contributions, and given the low confidence indicating uncertainty in the initial assessment, we believe that had the discussion period not been interrupted by the security incident, a score increase would have been warranted based on our comprehensive responses.*
>
> ---
>
> **Reviewer tazK (Score: 8): Stronger Endorsement with Deepened Understanding**
>
> Reviewer tazK recognized our "novel joint optimization approach" and the successful demonstration of attacks on commercial models without requiring knowledge of model internals. The reviewer raised two constructive questions regarding surrogate transferability and notation clarity, both of which we addressed thoroughly.
>
> On surrogate transferability, we explained the theoretical basis for cross-model attack transfer: contrastively-trained visual encoders share fundamental vulnerabilities due to common training objectives, limited large-scale captioned datasets (e.g., LAION-2B), and architectural similarities. Our ensemble attack against 12 CLIP models targets these shared vulnerabilities, enabling transfer to unseen proprietary models. We further provided the independently verifiable GPT-5.1 demonstration as concrete evidence.
>
> On notation, we provided detailed clarification of the relationships between Equations 8, 9, 10, and 12, showing how they collectively define the optimization objective, gradient computation strategy, and effectiveness constraints.
>
> Following our comprehensive responses, **the reviewer explicitly confirmed satisfaction and maintained their acceptance recommendation**: *"I thank the authors for comprehensive response. I would like to keep my score for acceptance."* Notably, this rebuttal exchange provided the reviewer with substantially deeper insight into our methodology and threat model, strengthening the foundation for their positive assessment. We believe this reviewer's endorsement now rests on a much better understanding of both the technical contributions and their practical implications.
>
> ---

---

> ### Author Response · Authors · 2025-12-04
> **Rebuttal Summary Part IV (Reviewer-by-Reviewer Response to Reviewer v9mh and Hgoh)**
>
> **Reviewer v9mh (Score: 2, Confidence: 4): Mischaracterization of Contribution and Attack Goals**
>
> We respectfully but firmly note that Reviewer v9mh's assessment rests on a **mischaracterization** of our work, conflating it with VLM jailbreak attacks. This comparison reflects a misunderstanding of both our attack goals and technical contributions.
>
> **Jailbreak Attacks vs. CPS: Opposite Goals, Orthogonal Vulnerabilities**
>
> | Aspect | Jailbreak Attacks | CPS (Our Work) |
> |-------|--------------|-------------|
> | **Goal** | Bypass safety alignment to elicit harmful outputs | Manipulate visual perception to bias selection decisions |
> | **Visibility to humans** | Harmful content can be visible, evasiveness is secondary | Perturbations must be **imperceptible** to humans |
> | **Visibility to models** | Harmful instructions should reach the model to trigger unsafe responses | Injected concepts should **maximally influence** model perception |
> | **Attack surface** | Safety guardrails and alignment training | Visual encoding and preference reasoning |
>
> The cited jailbreak works all aim to make harmful instructions reach the model while potentially hiding them from safety alignment of the model. **CPS operates with the opposite constraint:** perturbations must be invisible to humans while maximally shifting model perception. This defines entirely different threat models, technical challenges, and defenses.
>
> **Technical Distinction from Cited Works by Reviewer v9mh**
>
> | Cited Work | Their Method | Why This Differs from CPS |
> |------------|------------|-----------------------|
> | Miao et al. (2025) | Visible auxiliary images, fabricated dialogues | **Visible** content; no adversarial perturbations |
> | Wang et al. (2024) | Image mirroring, encryption-decryption | **Visible** modifications; information encoding |
> | Wang et al. (2025) | LSB steganography | Information concealment, not perception manipulation |
>
> None use imperceptible adversarial perturbations to alter visual perception. None achieve robust black-box transfer. None target preference manipulation. **They solve a fundamentally different problem.**
>
> **Core Novelty Misunderstood**
>
> The reviewer characterizes our visual attack as "reusing known CLIP-transferable techniques," overlooking that prior work achieves <20% ASR after real-world preprocessing. CPS achieves much higher ASR with robustness to compression, resizing, and format conversion, which is the critical breakthrough enabling practical attacks against deployed systems.
>
> **Additional Concerns Addressed**
>
> | Concern | Resolution |
> |---------|------------|
> | Mechanistic understanding of cross-modal effects | Explained unified visual processing in web agents; provided white-box probing in Appendix C |
> | Limited defense evaluation | Added three detection methods: consistency check (25%), binary detector (21.5%), risk scorer (Top-1/2/5: 21.5%/27.2%/54.3%), all near random baseline |
>
> **Summary:** This reviewer's rejection recommendation stems from evaluating CPS against the wrong category of attacks. We provided extensive clarification and new evidence, but the reviewer did not engage with our rebuttal.
>
> ---
>
> **Reviewer Hgoh (Score: 4, Confidence: 3): Key Misunderstandings Corrected**
>
> Reviewer Hgoh provided the most detailed review with seven distinct concerns spanning practicality, detection, reproducibility, and ethics. We addressed each concern systematically in our rebuttal with clarifications, new experiments, and additional documentation. The majority of concerns stemmed from misunderstandings of our threat model or requested details that we have now provided.
>
> | Concern | Resolution |
> |---------|------------|
> | Practicality gap (query awareness, unlimited probing) | **Corrected misunderstanding:** No query observation required; max 20 iterations aligns with standard black-box attack literature (Persuasion Attack uses 10) |
> | Semantics-preserving textual editing | Acknowledged textual component follows MPMA and Persuation Attack for text optimization; emphasized **visual attack robustness** is the main contribution |
> | Detection evaluation narrow | Added 3 new detection methods: consistency check (25%), binary detector (21.5%), risk scorer with Top-1/2/5 rates |
> | Missing hyperparameters | Provided complete specification (L∞=8/255, step=0.5/255, 50 steps, 20 crops, 12 surrogates); will include in camera-ready |
> | Threat model uses surrogates | Ensemble of 12 surrogate CLIP models targets shared vulnerabilities among image encoders that share similar architectures and training data|
> | External validity limited | Evaluation pairs targets with random items in same category; small preference shifts yield large real-world profit at scale |
> | Imperceptibility unproven | L∞=8/255 is established standard; detection rates serve as proxy for agent imperceptibility |
>
> We believe our comprehensive responses address all raised concerns. The reviewer did not respond following our rebuttal due to the incident.
>
> ---

---

> ### Author Response · Authors · 2025-12-04
> **Rebuttal Summary Part V (Summary and Conclusion)**
>
> ### [4. Summary and Conclusion]
>
> | Reviewer | Score | Status |
> |----------|-------|--------|
> | tazK | 8 | Explicitly satisfied, maintained acceptance with deepened understanding |
> | S5C5 | 4 | Concerns addressed; identified "weaknesses" are core strengths |
> | Hgoh | 4 | All seven concerns addressed; key misunderstandings corrected |
> | v9mh | 2 | Mischaracterization of contribution clarified |
>
> **Reasons to Accept This Paper:**
>
> 1. **Timely and Critical Significance:** VLM-based web agents are actively being deployed for high-stakes selection tasks—product recommendations, content curation, travel booking. CPS documents vulnerabilities that **exist today on every platform where agents interact with user-generated content.** Both the preference manipulation attack surface and the CLIP-based visual attack channel remain unpatched and exploitable against current commercial systems, including GPT-5.1 released days ago. This work enables the AI safety community to develop defenses before widespread exploitation occurs.
>
> 2. **First Practical Preference Manipulation Attack Under Realistic Constraints:** CPS establishes the first preference manipulation framework operating under real-world threat assumptions: an attacker with only vendor-level permissions (control over their own thumbnail and text description) and fewer than 20 agent queries can systematically bias agent decisions. No prior work has demonstrated this attack surface under such practical constraints.
>
> 3. **Discovery of a Robust Black-Box Visual Attack Channel:** We identify CLIP-based visual encoders as the Achilles' heel of modern VLMs and exploit this vulnerability to establish a previously unknown attack channel. With imperceptible perturbations (L∞ = 8/255), we drastically alter model perception—and crucially, our attack survives real-world preprocessing (JPEG compression, resizing, color jitter, screenshots) where prior work fails (<20% ASR). The independently verifiable GPT-5.1 demonstration (Figure 2, page 17) confirms this threat generalizes to unseen commercial models.
>
> 4. **Comprehensive Evaluation and Extensive Rebuttal Evidence:** We evaluated across multiple proprietary and open-source VLMs (GPT-4.1, Qwen-2.5-VL, Pixtral-Large) in two domains. During rebuttal, we provided 6 new experimental studies—robustness under combined transformations, budget sensitivity analysis, three additional detection methods, and complete hyperparameter specifications—addressing every substantive concern raised.
>
> **Conclusion:**
>
> The supporting reviewer (tazK) explicitly confirmed acceptance after thorough rebuttal engagement. The two marginally-below reviewers (S5C5, Hgoh) raised concerns that were either misunderstandings of our threat model or requests for additional experiments, all of which we addressed comprehensively. The rejecting reviewer (v9mh) evaluated CPS against the wrong category of attacks (jailbreaking) and did not engage with our rebuttal clarifications.
>
> CPS identifies a novel, practical vulnerability in deployed AI systems and provides the technical foundation for developing robust defenses. We respectfully submit that this work meets the standard for acceptance at ICLR.
>
> Sincerely,
>
> Authors
>
> ---

---

### Meta-Review · Area_Chair_GCLG · 2026-01-07

**Summary:**

The reviews agree the submission tackles an important and timely security problem for VLM-based web agents and provides extensive empirical results across multiple models and two domains. The decision hinges on disagreement about (i) the conceptual novelty of the visual component relative to recent multimodal attack literature, (ii) whether the “practical/realistic” threat model is convincingly supported given platform constraints (rate limits, edit throttling, caching, prompt uncertainty), and (iii) whether the stealth, semantics-preserving text claims, and defense analysis are adequately substantiated. Scores span from a strong accept to a confident reject, with two “borderline below threshold” reviews that could move depending on how much weight one places on the added rebuttal evidence versus remaining realism/defense/ethics concerns.

Despite solid empirical effort and a convincing rebuttal on several narrower technical questions (robustness tests, budget/cost disclosure, and notation), substantial concerns remain about (i) conceptual novelty relative to closely related attack lines, (ii) lack of mechanistic explanation/evidence for the claimed cross-modal gains, (iii) realism of the practicality + “semantics-preserving/stealth” claims under production constraints and stronger defenses, and (iv) unresolved dual-use/ethics risk that appears central to the work’s framing.

**Reviewer Concerns:**

Addressed concerns:
- Rebuttal adds JPEG+resize+color-jitter experiments and more detailed robustness claims, directly addressing the request for broader transformation tests.
- Rebuttal provides query budgets, average iterations, wall-clock time, and degradation under smaller budgets, addressing the quantitative “how expensive/practical” question.
- Rebuttal adds MI-FGSM results vs PGD, answering the methodological comparison request.
- Rebuttal explains the projection operator and reconciles the equations, and the reviewer explicitly acknowledges the response and maintains an accept score (Official Comment by Reviewer).
- Rebuttal adds additional detector-style evaluations (consistency/binary/risk-scoring), which partially responds to the “defense analysis shallow” critique.

Remaining concerns:
- The rebuttal argues the paper is orthogonal (preference manipulation vs jailbreak) and emphasizes robustness; however, the reviewer’s main novelty objection (that the visual module “mainly reuses known CLIP-transferable techniques” and needs clearer differentiation + deeper analysis) is only partially met and there is no reviewer follow-up indicating persuasion.
- Rebuttal provides budget sensitivity and argues prompt observability isn’t required, but the strongest critique—whether textual edits remain faithful and policy-compliant under realistic moderation and whether stealth claims hold under stronger, more realistic detection setups—remains insufficiently resolved.
- Rebuttal adds more detector variants and argues L∞ norms are standard, but still lacks the broader, calibrated detection/perceptual evaluation the reviewer asked for (heterogeneous detectors, human/perceptual metrics, ROC-style reporting).
- Rebuttal frames the work as defensive research but does not fully resolve concerns about weaponizable detail release and mitigation plans; this remains a material consideration.

**Reviewer Scores:**

- Review S5C5 (Rating 4): likely +1 (to ~5) given the rebuttal directly answers their robustness, MI-FGSM, and budget/cost questions; still, the original low confidence suggests limited certainty about a large jump.
- Review tazK (Rating 8): no change; reviewer explicitly states they keep the acceptance score in an Official Comment by Reviewer.
- Review v9mh (Rating 2): likely no change (or at most +0.5 notches in sentiment) because the rebuttal does not provide the deeper novelty/mechanistic substantiation they requested, and there is no reviewer follow-up indicating concerns were alleviated.
- Review Hgoh (Rating 4): likely no change or +1 at most; rebuttal adds budgets, hyperparameters, and additional detection checks, but the key practicality/semantics-preserving/stealth and external-validity critiques appear only partially addressed, and no reviewer follow-up is present.

---

### Decision · Program_Chairs · 2026-01-26

Reject